

# Monitoring the differential reflectivity and receiver calibration for the German polarimetric weather radar network

Michael Frech[1] and John Hubbert[2]

[1]Deutscher Wetterdienst, Observatorium Hohenpeißenberg, Albin-Schwaiger-Weg 10, 82383 Hohenpeißenberg
[2]National Center for Atmospheric Research, Boulder, Colorado

**Correspondence:** Michael Frech
Michael.Frech@dwd.de

**Abstract.** It is a challenge to calibrate differential reflectivity $Z_{DR}$ to within 0.1 - 0.2 dB uncertainty for dual-polarization weather radars that operate operationally 24/7 throughout the year. During operations, a temperature sensitivity of $Z_{DR}$ larger than 0.2 dB over a temperature range of 10 C has been noted. In order to understand the source of the observed $Z_{DR}$ temperature sensitivity, over 2000 dedicated solar box scans, a two dimensional scan 5° azimuth by 8° elevation that encompasses the

solar disk, have been made in 2018 from which horizontal (H) and vertical (V) pseudo antenna patterns are calculated. This assessment is carried out using data from the Hohenpeißenberg research radar which is identical to the 17 operational radar systems of the German Meteorological Service (Deutscher Wetterdienst, DWD). $Z_{DR}$ antenna patterns are calculated from the H and V patterns which reveal that the $Z_{DR}$ bias is temperature dependent changing about 0.2 dB over a 12°C temperature range. One-point calibration results, where a test signal is injected into the antenna crossguide coupler outside the receiver

box or into the LNAs, reveal only a very weak temperature sensitivity (<0.02 dB) for the receiver electronics. Thus, the observed temperature sensitivity is attributed to the antenna assembly. This is in agreement with NCAR's (National Center for Atmospheric Research) S-Pol (S-band polarimetric Radar) system, where the primary $Z_{DR}$ temperature sensitivity is also related to the antenna assembly (Hubbert 2017). Solar power measurements from a Canadian calibration observatory are used to compute the antenna gain, and to validate the results with the operational DWD monitoring results. The derived gain values

agree very well with the gain estimate of the antenna manufacturer. The antenna gain shows a quasi-linear dependence with temperature with different slopes for the H and V channels. There is a 0.6 dB decrease in gain for a 10°C temperature increase, which directly relates to a bias of the radar reflectivity factor Z which has not been not accounted for previously. The operational methods to monitor and calibrate $Z_{DR}$ used for the polarimetric DWD C-band weather radar network are discussed. The prime sources to calibrate and monitor $Z_{DR}$ are birdbath scans, which are executed every 5 minutes, and the analysis of solar spikes

that occur during operational scanning. Using an automated $Z_{DR}$ calibration procedure on a diurnal timescale, we are able to keep $Z_{DR}$ bias within the target uncertainty of $\pm 0.1$ dB. This is demonstrated for data from the DWD radar network comprising over 87 years of cumulative dual-polarization radar operations.



# 1 Introduction

Dual-polarization (dualpol) weather radars have become the standard in European weather radar networks. Typically national weather service radars operate in the STAR mode (simultaneous H and V transmit and receive H and V), sometimes referred to as SHV (simultaneous H and V) mode. Polarization moments, such as differential reflectivity ($Z_{DR}$) can be used to better characterize the hydrometeors (e.g. Straka et al., 2000, Schuur et al., 2012, Al-Sakka et al., 2013, Steinert et al., 2013) and to better quantify the precipitation amount (e.g. Bringi and Chandrasekar, 2001, Ryzhkov et al., 2005, Bringi et al., 2011, Diederich et al., 2015). Dualpol moments are also used to improve the data quality control via fuzzy logic clutter classifiers (Werner and Steinert, 2012, Hubbert et al. 2009a,b).

In order to keep biases in quantitative precipitation estimates under 20%, $Z_{DR}$ should be calibrated to an accuracy better than $\pm 0.2$ dB (Bringi and Chandrasekar, 2001), assuming no bias in reflectivity ($Z$). In order to quantify the bias in $Z_{DR}$, the differential gain/loss of both the H and V transmit and receive paths needs to be assessed as well as the differential transmit power. Active components in the H and V receive paths, such as the LNAs (low noise amplifiers) are never perfectly matched, are temperature dependent and thus are a possible source of time varying $Z_{DR}$ bias. The differential gain of entire signal path has to be quantified and removed from the measured $Z_{DR}$ in order to obtain an accurate estimate of intrinsic $Z_{DR}$. The most well established way to calibrate $Z_{DR}$ is via vertical pointing scans in light rain (Gorgucci et al. 1999, Bringi and Chandrasekar 2001). The premise is that the rain particles are polarimetrically isotropic when viewed vertically so that $Z_{DR}$ is 0 dB. This technique works well since it is a "end-to-end" measurement that takes into account both the transmit and receive paths and the radar resolution volume is filled with distributed scatterers. Another technique to calibrate $Z_{DR}$ is to characterize the transmit and receive paths of the radar with RF sources and power meters (Hubbert et al., 2008, Zrnic et al. 2006). However, it has been found that such estimates can have large uncertainty and thus research radars such as CSU-CHILL and S-Pol employ the vertical pointing technique to attain a reliable $Z_{DR}$ calibration. The engineering technique has also proved untenable for the Next-Generation Radars (NEXRADs) (Ice et al. 2014).

In Hubbert (2017) the cross-polar power technique for $Z_{DR}$ calibration is applied to data from NCAR's dual-polarimetric S-band radar, S-Pol. In contrast to most operational radars, S-Pol uses a fast switch to alternate between H and V only transmit polarizations on a pulse-to-pulse basis. Both H and V polarizations are received thus providing measurements of the crosspolar signal, which is not measured in STAR mode. S-Pol is operated without a radome and the receiver is located in a container which is temperature controlled. The DWD radars have antenna mounted receiver electronics and operate within a radome. In Hubbert (2017) systematic $Z_{DR}$ temperature dependence was found using an analysis of solar scan data, crosspolar measurements and transmit power monitoring. It was shown that the temperature dependent gain of antenna assembly caused the observed $Z_{DR}$ biases. Frech et al. (2013) also investigate the characteristics of a DWD antenna in part using solar measurements. The antenna characteristics for H and V polarizations must match very well, not only during the acceptance of a system, but also during subsequent day-to-day operations. For example, because of mechanical stress over time, the feed horn could defocus which may result in increased side-lobe levels and increased beam-squint. This in turn would affect the clutter suppres-





sion performance and the interpretation of $Z_{DR}$ in areas with large reflectivity gradients. Furthermore, antenna cross coupling must be small in order to avoid additional $Z_{DR}$ bias (Wang et al., 2006, Hubbert et al. 2010, Zrnic et al., 2010)

In order to both calibrate and monitor the $Z_{DR}$ bias, DWD radars employ a vertical pointing scan (sometimes called bird bath scans) executed every 5 minutes. It has been found that it is necessary to make at least one full 360° azimuth sweep when

pointing vertically in order to eliminate azimuth dependent effects on $Z_{DR}$ (Gorgucci 1999, Bringi and Chandrasekar, 2001). DWD has successfully applied this method in rain, mixed phase and solid phase precipitation. For systems that are not able scan at 90° elevation angle, $Z_{DR}$ bias is evaluated from specific precipitation situations. Evaluation of $Z_{DR}$ in Bragg-scatter areas is another potential $Z_{DR}$ monitoring method which is, however, mainly suitable for S-band systems (e.g. Richardson et al., 2017).

$Z_{DR}$ monitoring methods that use solar radiation are now commonly employed in operational weather radar networks (Holleman et al. (2010), Huuskonen and Holleman (2007), Figuras et al., 2013, Frech 2013, Huuskonen et al., 2016, Frech et al.). The sun can be considered as an unpolarized source of S-band radiation (i.e., the H and V powers are equal). Solar radiation can also be used to calibrate the receive path gain as well as the navigation position of a radar system. These techniques use solar spikes that are observed during normal operational scanning and thus can be continuously done without interrupting the

radar operations. Though the solar method only calibrates the receive path of a radar system, this method is considered as an essential element to monitor $Z_{DR}$. It is complementary to the birdbath method which relies on the presence of precipitation above the radar site.

In this paper the $Z_{DR}$ monitoring methods that are employed across the DWD weather radar network, which consists of 17 radar systems, are described. In the course of operating this radar network since 2009, a $Z_{DR}$ temperature dependence has

been found and documented (Frech, 2013). A goal of this paper is to identify the source of this $Z_{DR}$ temperature dependence. Similar to Hubbert (2017), solar scans are employed for a systematic analysis using the Hohenpeißenberg research radar (Frech et al., 2017). Each solar scan takes about 4 minutes and is repeated every 10 minutes. Up to 90 scans are available to assess the diurnal $Z_{DR}$ variability due to temperature. The pseudo-$Z_{DR}$ antenna patterns based on the solar scans are compared to $Z_{DR}$ antenna pattern measured during a dedicated antenna pattern measurement (Frech et al., 2013). Antenna beam widths

derived from the solar scans are compared to the beam widths measured during the antenna pattern measurements. The diurnal variation of solar differential power $S$ is used to assess the operational $Z_{DR}$ monitoring results from birdbath scans and $S$ measurements derived from solar interferences extracted from operational data. This analysis is complemented with results from continuous one-point calibration data where a test signal is injected in either the antenna coupler (before the TR-limiter) or just before the low-noise-amplifier (LNA) by using a built-in test signal generator (TSG).

The $S$ temperature dependence is also investigated in terms of antenna gain which is determined from solar power measurements at C-band (Sirman and Urell, 2001). This also provides an insight on how well the two receiver chains are calibrated. Those gain estimates are compared to four operational radar sites where one full diurnal cycle of solar boxscans was acquired, respectively. The performance of the operational $Z_{DR}$ calibration of the radar network based on birdbath measurements is discussed. This analysis is based on a combined 87 years of radar operation. In addition, based on the operational monitoring,

an example for an unusual failure of a TR-limiter is shown. The main findings are summarized in the conclusions.





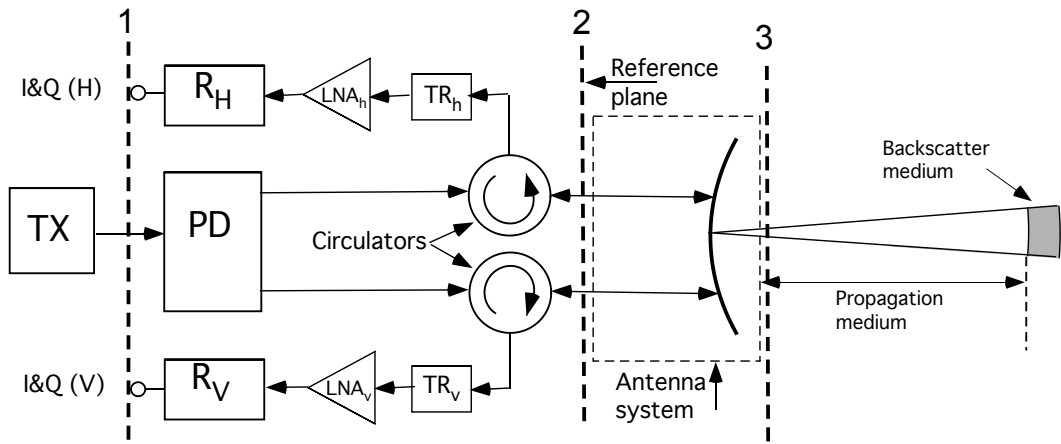

**Figure 1.** A general block diagram of the DWD radar systems. TX refers to the transmitter, PD is the power divider and $TR_{h,v}$ are the TR-limiters in the H & V receive path. I&Q denote the plane where the received analog signal at the intermediate frequency (IF) is digitized in the IFD (intermediate frequency digitizer).

## 2 Operational $Z_{DR}$ adjustment

Given in Fig. 1 is a radar block diagram that capturers the essential components affecting $Z_{DR}$. The vertical dashed lines mark measurement planes that are useful for defining the various gains and powers of the radar.

$$Z_{DR}^{bias} = \frac{TX_H\,(G_H^A)^2\,RX_H}{TX_V\,(G_H^A)^2\,RX_V} \tag{1}$$

5    where, $TX_{H,V}$ are the transmit powers measured at plane 2, $G_{H,V}^A$ are the H and V antenna gains, $RX_{H,V}$ are the H and V receiver gains from plane 2 to plane 1. The differential gains are defined as,

$$\Delta_{TX} = \frac{TX_H}{TX_V} \tag{2}$$

$$\Delta_{RX} = \frac{RX_H}{RX_V} \tag{3}$$

$$\Delta_A = \frac{G_H^A}{G_V^A} \tag{4}$$

10    so that the $Z_{DR}$ bias (also referred to as the $Z_{DR}$ offset) can be written as

$$Z_{DR}^{bias} = \Delta_{TX,RX} = \Delta_{TX}\Delta_{RX}\Delta_A^2 \tag{5}$$





in linear form, and in dB,

$$Z_{DR}^{bias} = \Delta_{TX,RX} = \Delta_{TX} + \Delta_{RX} + \Delta_A^2 \tag{6}$$

For DWD radars, the prime source to determine the $Z_{DR}$ offset is the birdbath scan. The key assumption of the method is that $Z_{DR}$ is zero when looking vertically at falling precipitation. Azimuthal dependent effects are accounted for by averaging

$Z_{DR}$ data over a full rotation of the antenna, i.e., $360°$ azimuth (Bringi and Chandrasekar 2001). A deviation from zero dB is then attributed to a $Z_{DR}$ offset or bias. The differential bias in the $\Delta_{RX}$ path can be due to differential receiver gain, circulator attenuation, LNAs and other electronics in H and V electronic paths. Offsets in $\Delta_{TX}$ is primarily a function of the power divider circuity and the circulators. All of these components have temperature dependent gains. The differential offset due to $\Delta_A$ are due to uncertainties in the antenna characterization (H & V antenna gain, width of the main lobe) and antenna

temperature (we will discuss this later).

$Z_{DR}$ bias estimated from a birdbath scan is computed as a range averaged $Z_{DR}$ in the antenna far-field (starting at about 700 m range). Threshold requirement are, $\rho_{hv} > 0.9$ (copolar correlation coefficient, Bringi and Chandrasekar 2000) and SQI $> 0.5$ (Signal Quality Index or normalized coherent power[1]), and at least ten valid range bins in a ray. Then the median $Z_{DR}$ is computed from all rays of the sweep. In order to obtain the diurnal averaged $Z_{DR}$, the median $Z_{DR}$ from all birdbath scans of

a given day is computed. A median $Z_{DR}$ is computed only if there are at least 6 birdbath scans available with valid data. There is no further separation according the hydrometers as the method is also applicable to mixed phase and solid phase precipitation (see e.g. Dixon et al, 2018).

The automated procedure to adjust the $Z_{DR}$ offset on a diurnal bias has been implemented. If the offset $\Delta_{TX,RX}$ is properly set initially at a given day, we have

$$0 \quad = \quad ZDR_{90°,t_i} - \Delta_{TX,RX}, \tag{7}$$

where $\Delta_{TX,RX,t_0}$ is the static offset which is stored as an initial parameter in the signal processor and $ZDR_{90°,t_i}$ is the current diurnal $Z_{DR}$ value determined from the birdbath scan. $\Delta_{TX,RX}$ is a value that has been determined during a routine maintenance.

If the following is found at a given day $t_i$:

$$0 \quad \neq \quad ZDR_{90°,t_i} - \Delta_{TX,RX} \tag{8}$$

then the offset needs to be adjusted. The nonzero $Z_{DR}$ offset is now

$$ZDR_{TX,RX,t_i} \quad = \quad ZDR_{90°,t_i} - \Delta_{TX,RX} \tag{9}$$

In order to calibrate $Z_{DR}$, the initial $Z_{DR}$ offset is corrected. The new system offset $\Delta_{TX,RX,t_i}$ is then

$$\Delta_{TX,RX,t_i} \quad = \quad \Delta_{TX,RX} + ZDR_{TX,RX,t_i} \tag{10}$$

---

[1]Defined as as the ratio of the autocorrelation function at lag-1 to lag-0.





The correct $Z_{DR}$ offset is stored in the metadata set of every ODIM HDF5 sweep that is sent to the central DWD radar data processing site. As part of the centralized quality control, this offset is applied to the $Z_{DR}$ data prior any product generation. The corrected offset is valid until a new $Z_{DR}$ offset can be computed from birdbath data. Depending on the season, weeks may pass until the next precipitation event occurs that can be used to assess the $Z_{DR}$ offset. In principle there could be drifts in the

$Z_{DR}$ bias which would introduce a $Z_{DR}$ bias during a dry period. However our experience indicates that the radar hardware state is sufficiently stable and precipitation is frequent enough so that the mean $Z_{DR}$ bias over a day would be smaller 0.1 dB in these dry periods. This is further discussed in section 11.

In order to assess the differential power bias in the entire receive path, the solar measurements can be used,

$$S = \frac{S_H}{S_V} = \frac{G_H^A \, RX_H}{G_H^A \, RX_V} \tag{11}$$

where $S_{H,V}$ are H and V solar powers estimated from scanning the sun. Since the sun is an unpolarized source of radiation, intrinsic differential solar power, $S$ should be zero dB. Thus, a measurement of non-zero differential solar power indicates a $Z_{DR}$ offset caused by differential losses/gains in the receive path. Typically, the radome attenuation is specified by the manufacturer and the antenna gain in H and V are often measured once on a test range prior to the onsite radar installation. Onsite antenna pattern measurements were made to verify the antenna specifications after the Hohenpeißenberg radar installation (Frech et al.,

2013). Antenna pattern measurements with and without the radome verified the antenna specifications. First sidelobe levels are within 0.5 dB of the specification and the antenna main beam width is within 0.03° of the specification.

## 3   Solar box scans

In this section the configuration and set-up of the solar box scans is described. The scanning azimuth and elevation limits of a box scan is of 8° by 5°, respectively, centered on the position of the sun at the beginning of the scan. The scan takes about

4 minutes to complete and is scheduled every 10 minutes. During this time the scan box azimuth and elevation limits are not adjusted to account for the movement of the sun, however, in the post collection analysis of the data, the movement of the sun is corrected by using the ray time tag from which the position of the sun is determined. The radar operates with a pulse length of 0.8 $\mu s$ (0.4 $\mu s$), with a scan rate of $2°\mathrm{s}^{-1}$ ($0.1°\mathrm{s}^{-1}$) with a PRF of 800 Hz (1600 Hz). The elevation angle increment is 0.1°. Range resolution is set to be 250 m and data up to a range of 150 km are acquired.

The system is transmitting while scanning and the PRF is chosen such that the system is operated with a constant duty cycle. It is important that the transmitter operates while making solar scans so that the radar components of the transmit and receiver path are in an operational state. There are three circulators built in the system which prevent reverse power from damaging the system. One of the circulators is located just behind the magnetron. The forward power measurements are taken through a coupler behind the circulator. The measured forward power decreases by 0.3 dB after a cold start until it reaches steady state.

Therefore, in order to mimic operational conditions, the transmitter is operating while doing the boxscans.

The integrated solar powers are corrected for noise. Prior to each solar scan, the thermal background noise is estimated at the elevation of the sun and about 30° off the azimuth of the sun. Overcast situations do not bias the solar scans. The radar moments





for each boxscan are saved in a HDF5 format. In the analysis the signal-to-noise ratio SNRh and SNRv (horizontal and vertical, respectively), the cross correlation coefficient $\rho_{hv}$, the differential solar power $S$ (stored as "$Z_{DR}$") are used. All data moments are computed from unfiltered time series (no Doppler clutter filter is applied). To avoid ground clutter contamination, only data beyond 50 km range are used, however, at low elevation angles data beyond 50 km range can be contaminated by clutter. If this

is the case, those data are removed from the analysis.

The standard lightning protection of a radar system consists of four vertical lightning poles in the vicinity of the radome which exends above the highest point of the radome. For the Hohenpeißenberg radar system, prior the measurements used in this study, three of the four lightning poles were removed in oder to avoid disturbances on radar data due to the lightning protection hardware.

## 3.1   Solar data analysis

Solar boxscan data analysis is described in this section. The analysis of solar box scans employs the methods that are used to evaluate solar interferences (sun spikes) from operational scans (Huuskonen and Holleman, 2007, Frech, 2013). The methods are extended to compute the antenna beam width from the solar scan (Huuskonen et al, 2014). The computed antenna beam width calculated from a solar box scan is a proxy since the solar disk (0.53°) is convolved with the antenna pattern (0.9° beam

width) measurement, and thus the observed solar disk and antenna pattern is smeared in azimuth and elevation. However, the beam width estimated from the solar scan is very close to the classic antenna pattern measurements (Frech et al. 2013). The results are shown in the next section. The positioning error and beam squint are also computed from this approach. The results related to positioning error are discussed in a companion paper (Frech et al., 2019). In order to determine the differential solar power bias of the receiver chain, the differential solar power is integrated over a 1° solid angle (± 0.5° relative to the beam

center).

The calibration of the receiver (dBm0) can be verified by comparing the measured solar power with independent solar power measurements. The solar flux measurements are available daily (2-3 times) from the Dominion Radio Astrophysical Observatory (DRAO) in Canada (Tapping, 2001). It is a solar flux measurement monitored at a wave length of $\lambda = 10.7$ cm (S band) with an expected accuracy of 1 sfu (Solar Flux Unit) (Tapping, 2013). This corresponds to a 0.02 dBm accuracy of the

power measurement. This is the independent flux measurement which is used to monitor the absolute receiver sensitivity of our radar system. As a first step, the S-band solar flux has to be converted to the corresponding C-band flux. Parameterizations of solar C-band flux as a function of the S-band flux are documented in literature (e.g. Holleman et al, 2010): .

$$F_C = 0.71 \left( F_{10.7} - 64 \right) + 126 \qquad (12)$$

with $F_{10.7}$, the adjusted solar flux (in sfu) from DRAO (a sfu has units of $10^{-22} W \cdot \left( m^2 Hz \right)^{-1}$). The maximum measured

received solar power needs to take into account the receiver bandwidth $\Delta f$ and the effective antenna area $A_e$. $A_e$ is defined as $A_e = \eta \cdot A = 0.55 \cdot \pi \left( 0.5 \cdot d_e \right)^2 = 7.876$ m$^2$ where an antenna efficiency of $\eta = 0.55$ is assumed (Sirmans and Urell, 2001). The electrical diameter of the antenna $d_e$ is 4.27 m.





The solar power at C-band $p_{sun}$ in $mW$ can be written as:

$$p_{sun} = \frac{1}{2} \cdot 10^{-13} \cdot \Delta f \cdot A_e \cdot F_C \tag{13}$$

For the C-band system at Hohenpeißenberg the receiver bandwidth $\Delta f \approx 2$ MHz (for short pulse, $0.8\mu s$; for long pulse $2$ $\mu s$ $\Delta f \approx 0.6$ MHz ). The factor of 0.5 is introduced since the solar flux is an unpolarized source whereas the radar system receives
power at horizontal or vertical polarization.

The solar power is determined from the measured peak SNR that is estimated from the solar scan.

The solar power in H and V can be written as

$$P_{h,v} \quad = \quad SNR_{h,v} + dBm0_{h,v} + A_{gas} + k \tag{14}$$

where $A_{gas}$ denotes the gaseous attenuation, $dBm0_{h,v}$ is the minimum detectable power of the receiver, and $k$ the beamwidth
correction factor. For the Hohenpeißenberg radar the minimum detectable power is $dBm0_h = -109.54$ dBm and $dBm0_v = -109.42$ dBm for a pulse length of 0.8 $\mu s$. Those power levels are determined as part of the engineering radar calibration.

Before the measured solar power $P_{h,v}$ can be related to the received solar power, a correction for the one-way gas attenuation $A_{gas}$ of the solar power due to the earths atmosphere has to be applied. This is estimated using a 4/3 earths radius model where the ray path $r$ up to the top of the atmosphere is approximated using a standard atmosphere (Holleman et al, 2010):

$$r(z, el) = R_{43}\sqrt{\sin^2 el + \frac{2z}{R_{43}} + \frac{z^2}{R_{43}^2}} - R_{43}\sin el \tag{15}$$

The gaseous attenuation can be approximated as:

$$A_{gas}(el) \approx a \cdot r(z_0, el) \tag{16}$$

with $z_0$, the equivalent height of a homogeneous atmosphere. A homogeneous atmosphere is defined by constant air density with height. Using typical values of a standard atmosphere, the height of a homogeneous atmosphere is $z_0 \approx 8.4 km$. For $a$ we
assume $a = 0.008$ dB$\cdot km^{-1}$.

The peak solar SNR determined from the boxscan requires a beam width correction factor $k$ because the solar disc is smaller than antenna beam width (Sirmans and Urell, 2001):

$$k = \left[1 + 0.18\left(\frac{\theta_s}{\theta_{3dB}}\right)^2\right]^2 \tag{17}$$

with $\theta_s = 0.57°$ and $\theta_{3dB} \approx 0.9°$. This yields $k = 1.14$ dB.
From the computed solar powers a differential power $S$ is defined as:

$$S \quad = \quad P_h - P_v. \tag{18}$$





All moments and power estimates at a ranges larger 50 km are averaged and the results given in the next sections show range averaged data.

One way to compare the solar power measurements from DRAO with radar estimated solar power is to compute the antenna gain (Sirmans and Urell, 2001). This has the advantage to easily compare results from different pulse widths and different radar systems. The 3 dB beam width, the geometric and electric antenna dimension are known. Furthermore, a constant antenna efficiency is assumed when computing $A_e$. The antenna gain is defined as (e.g. Sirmans and Urell, 2001)

$$g = \frac{4\pi A_e}{\lambda^2} \tag{19}$$

with the radar wave lenght $\lambda$. Using eq. 13 we can write this as

$$g = \frac{4\pi}{\lambda^2} \left( \frac{p_{sun} \cdot 2}{10^{-13} \cdot \Delta f \cdot F_C} \right) \tag{20}$$

We compute two gain values. First using the independent data from DRAO to compute $p_{sun}$, and secondly computing $p_{sun}$ using Eq(14). A difference between those two gain estimates using the independent solar power measurements at C-band and the radar measured solar power can be interpreted as a receiver calibration bias, and thus a bias of $dBm0_{h,v}$. The antenna efficiency $\eta$ is fixed in both bain estimates. The advantage to use gain as a retrieval parameter instead of the solar flux (like it is commonly done in literature, where the radar measured received solar power is converted into solar flux units) is that the time variability of the solar flux is removed. So differences in gain estimates using solar power measured by radar and the independent gain estimate based on DRAO data provide a straight forward information on a relative receiver calibration bias. Since basic antenna parameters are fixed, the gain estimates cannot be viewed as true antenna gain estimates.

## 4 Analysis of pseudo $Z_{DR}$ antenna patterns

The typical solar beam plot for the SNRh and SNRv pattern are shown in Fig. 2. The corresponding differential solar power $S$ pattern and the cross correlation coefficient $\rho_{hv}$ is shown in Fig. 3.

There are four areas of large differential powers at a radii of $1°$ (Fig. 3). This differential solar power $S$ pattern is supported by analyzing the 3dB beamwidth employing the method of Huuskonen et al. (2014). The timeseries of 3dB beamwidths from 91 solar box scans are shown in Fig. 4 and the corresponding azimuth and elevation of sun during that day are shown in Fig. 5. We show the series in order to illustrate the the consistency of the results throughout a day for different elevations and azimuths. The mean azimuth width is $0.94° \pm 0.01°$ and $0.98° \pm 0.01°$ for H and V, respectively. The elevation width in H and V is $0.95° \pm 0.006°$ and $0.89° \pm 0.006°$, respectively.

The beamwidth results indicate a near circular beam shape in H, and in contrast, the beam shape is more elliptical in V. The superposition of the circular and elliptical beam shape leads to the observed solar "$Z_{DR}$" or $S$ pattern.

During the Hohenpeißenberg acceptance tests a series of antenna pattern measurements were carried out (Frech et al., 2013). An example $S$ pattern is shown in Fig. 6. The dynamic range of an antenna pattern measurement is of course much larger (peak



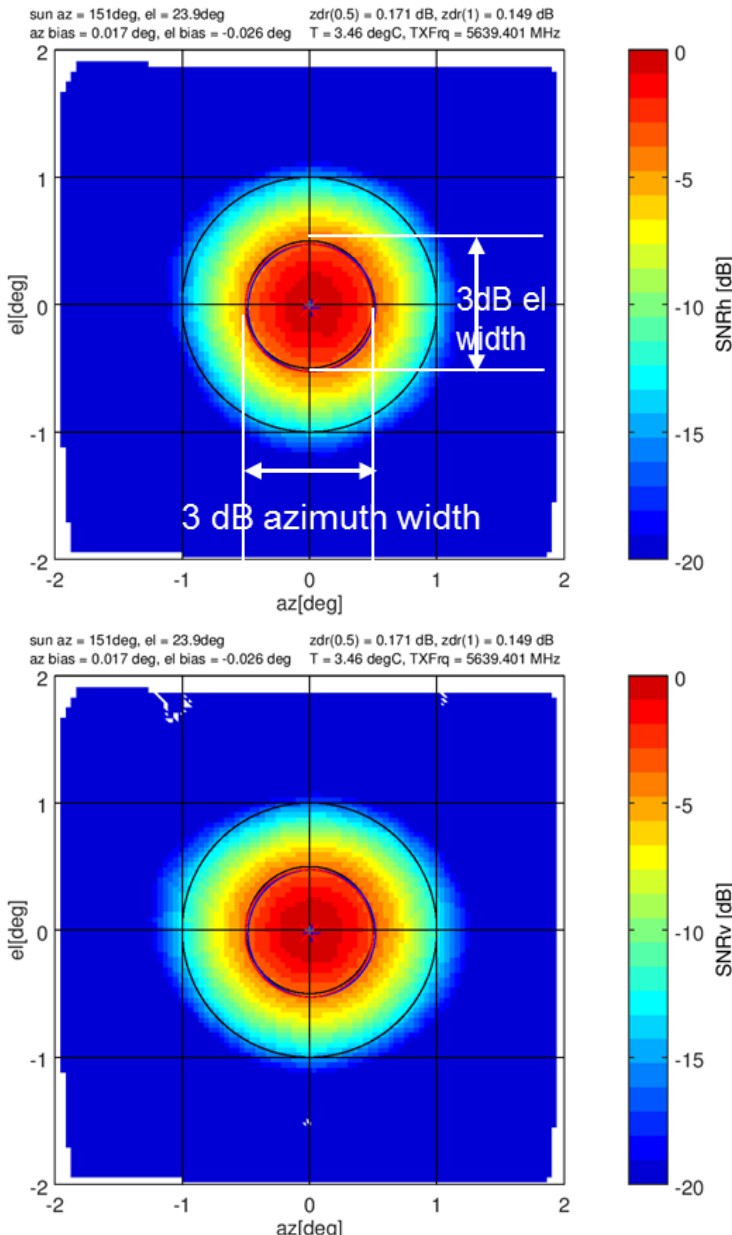

**Figure 2.** Typical SNR solar beam plot from a boxscan, SNRh, upper panel, and SNRv, lower panel. This plot illustrates how the solar beam widths are computed. On top of each panel, the corresponding position of the sun relative to the radar, the computed elevation and azimuth radar positioning bias, the transmit frequency, the radome temperature and the $Z_{DR}$ integrated over 0.5 and 1° radii is given as a standard information data set with this product.



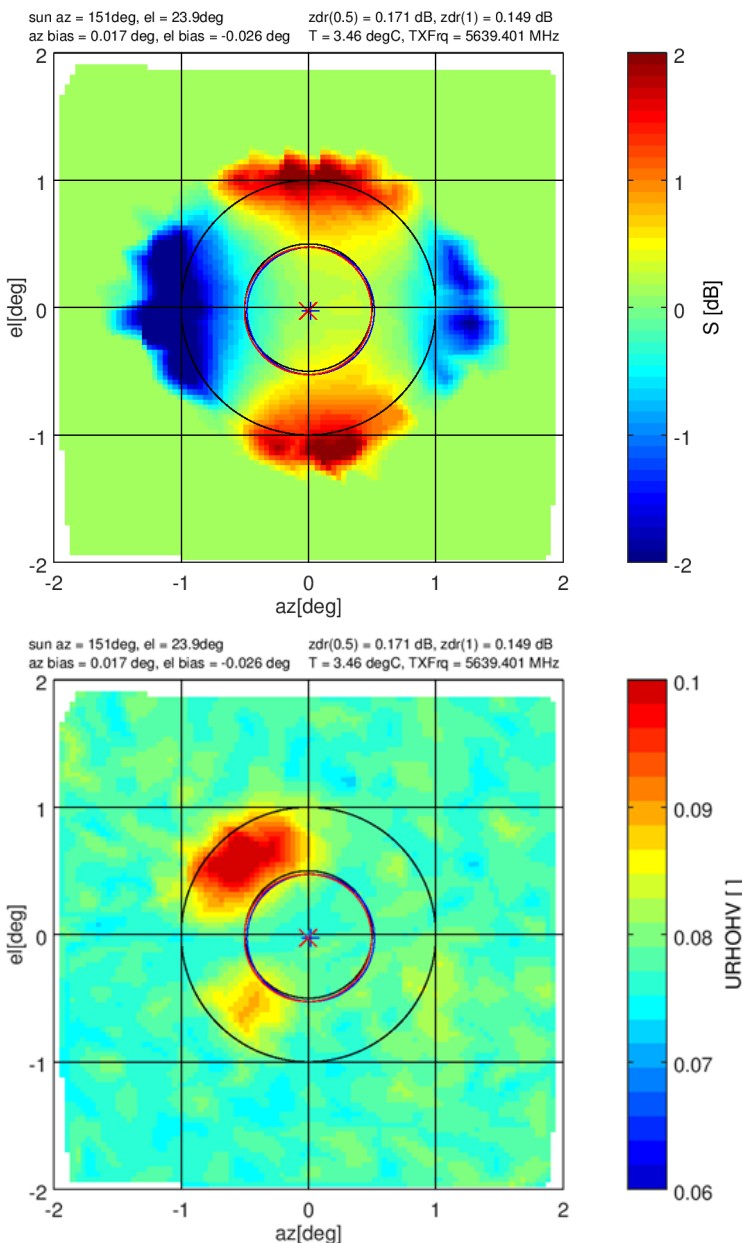

**Figure 3.** The corresponding pattern of differential solar power $S$ is shown in the upper panel (to be compared with Figure 2). The cross channel correlation coefficient $\rho_{hv}^{S}$ is shown in the lower panel. Note the narrow scale from 0 to 0.1.





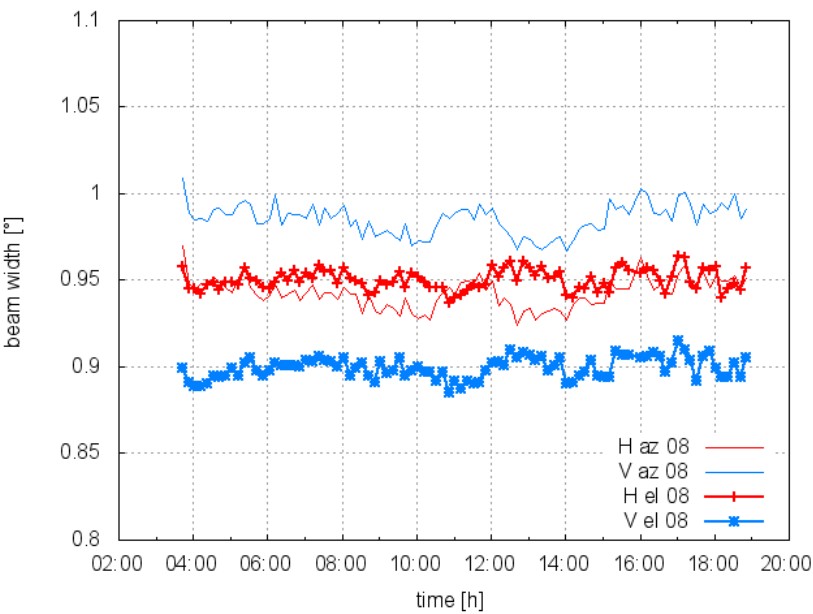

**Figure 4.** Timeseries of 3 dB beam widths from 91 solar boxscans. Data are taken on 23 June 2018. Shown are the beam widths in vertical and horizontal dimension, and for horizontal and vertical polarization.

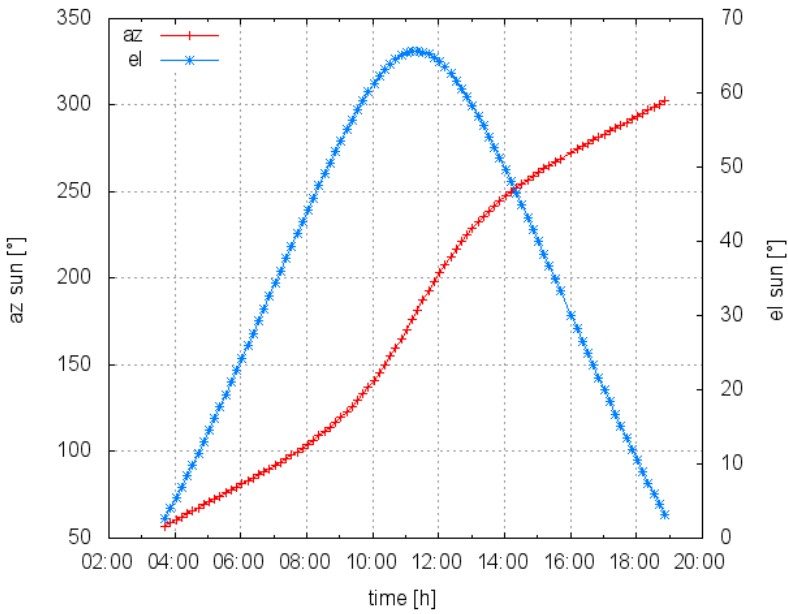

**Figure 5.** The diurnal cycle of elevation and azimuth position of the sun (23 June 2018) which corresponds to the solar boxscans that are used to compute the beam widths in Fig. 4.





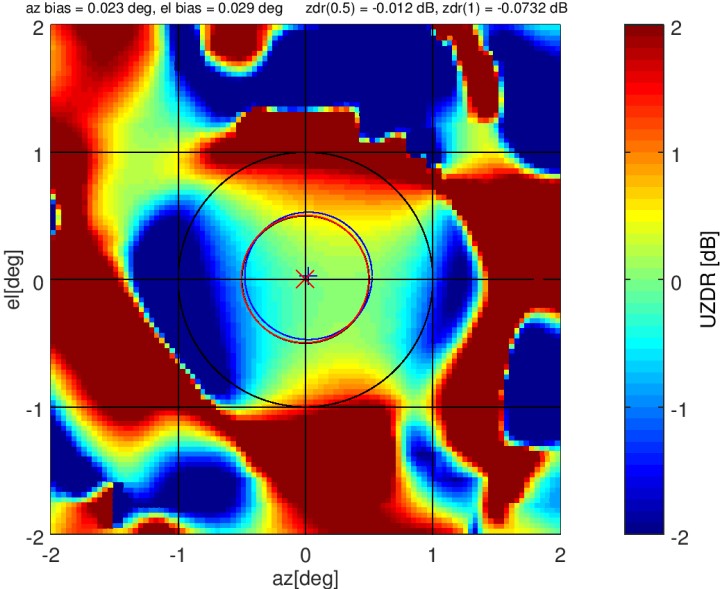

**Figure 6.** Hohenpeißenberg $Z_{DR}$ antenna pattern taken at 18 May 2011, to be compared with the pattern of differential solar power (Fig. 3, upper panel).

SNR of the external source is 68 dB compared to the peak solar SNR of about 7 dB) so that there is differential power visible outside the main beam. But within the main beam ($\pm 1°$) the $S$ patterns show a remarkable agreement. Thus, the mainlobe of solar $S$ patterns can be used to assess the antenna performance without carrying out dedicated antenna pattern measurements. Taking such a measurement on a regular basis throughout the lifetime of a radar system ($\approx 20$ years) helps to monitor the state

of the antenna assembly.

The complex H and V time-series data resulting from scanning the solar disk can also be used to create a cross-channel correlation antenna pattern. The simultaneous received voltage time-series from a single dwell angle, $V_h(i)$ and $V_v(i)$, for the horizontal and vertical channels, respectively, are correlated as

$$\rho_{hv}^S = \frac{\sum_{i=1}^N V_h(i)V_v^*(i)}{\sqrt{\sum_{i=1}^N V_h(i)V_h^*(i) \sum_{i=1}^N V_v(i)V_v^*(i)}} \tag{21}$$

where N is the number of samples. Thus $\rho_{hv}^S$ gives the point-wise (spatial) correlation from temporal averages. This correlation data from all dwell angles is interpolated to a grid. The resulting magnitude of the correlation product of Eq.(21) is shown in the lower panel of Fig. 3. If solar radiation is unpolarized, the correlation of data between any two orthogonal receive polarization channels is zero by definition. The correlation magnitude in Fig. 3 (bottom) shows two principal "lobes" in the left two quadrants where the correlation increases. These two areas of increased correlation coefficient are manifestations of

the antenna polarization errors (Hubbert et al. 2010a,b). This signature is present throughout nearly all of the solar boxscan measurements.



## 5 Differential solar power $S$ time series based on box scans and comparison to the operational $S$ and $Z_{DR}$ monitoring

Over 2157 solar boxscans were made in order to study the variability of $S$ . Before a statistical analysis is performed on this data, two illustrative $S$ time series from two particular days are shown. The example from 3 June 2018 is shown in Fig. 7. The

mean differential solar power $S$ is -0.19 dB and over the day the standard deviation is 0.032 dB. The standard deviation was determined after the removal of the trend using a 5th order polynomial fit to the data. In addition two temperature measurements are shown. One is termed as the radome temperature which indicates the temperature in the radome to which the antenna assembly is exposed. The LNA temperature (low-noise amplifier) is a temperature reading close to the LNA's within the receiver box which is assumed to be representative for the temperature condition within the receiver box. By eye there seems

to be a correlation between temperature and $S$. From similar data gathered on 15 June 2018 in Fig. 8, this correlation appears more obvious. $S$ increases by about 0.2 dB over this day with a temperature increase of about 7°C. For this case the mean $S$ is -0.15 dB and the standard deviation is 0.038 dB. In principle we would expect a constant $S$ throughout the day because the sun is an unpolarized source of radiation. So besides the obvious temperature dependence, the variability of $S$ (expressed in terms of the standard deviation) may be caused by the radome, insufficient cross-polar isolation of the antenna and/or clutter, or just

random sampling errors. There are no means to separate those effects with the existing measurements.

Next, the time series of $S^2$ are compared to the operational monitoring results of differential solar power $S^2$, which are based on solar interferences identified from the operational scanning data, and to $Z_{DR}$ data from birdbath scans ($S$ is squared to make it comparable to $Z_{DR}$). In order to compare $S$ from boxscans to the operational results, the system $Z_{DR}$ offset of $-0.1$dB is subtracted from those data. There is only one $S^2$ and $Z_{DR}$ value per day from operational monitoring, which is indicated by

the two straight lines in Figs. 9 and 10, respectively. $Z_{DR}$ from the birdbath scans is only updated if there has been sufficient precipitation on the previous day. Otherwise $Z_{DR}$ of the last precipitation event is assumed to be still valid.

There is on average a very good agreement between the $S^2$ derived from operational scanning and compared to the $S^2$ from the box scans (Figs. 9 and 10). It is obvious that diurnal temperature variations cannot be captured by $S^2$ which is derived from operational scanning. The respective $Z_{DR}$ from the birdbath scan on 3 June is near 0 dB compared to -0.1 dB of $S^2$ . The

opposite is found on 15 June where the birdbath $Z_{DR}$ is -0.1 dB and $S^2$ is near 0 dB.

## 6 Antenna gain based on solar power measurements

The antenna gain is computed using the measured solar power by the radar and the DRAO solar flux (see Eq.(20)). If the receive path is properly characterized and calibrated, the retrieved antenna gain should be the antenna gain as provided by the antenna manufacturer. If the solar power based on the DRAO is used to compute the antenna gain using Eq.(20), a deviation

from the gain value provided by the manufacturer is due to the assumption on the antenna characteristics. Similarly, by using the radar measured solar power, a systematic deviation from the manufacturer gain is indicative of a calibration error.

The antenna gain results are shown for the same dates as before in Figs. 11 and 12 where the timeseries of gain for both polarizations are shown as function of the solar azimuth position. The antenna gain results (about 90 estimates based on 90





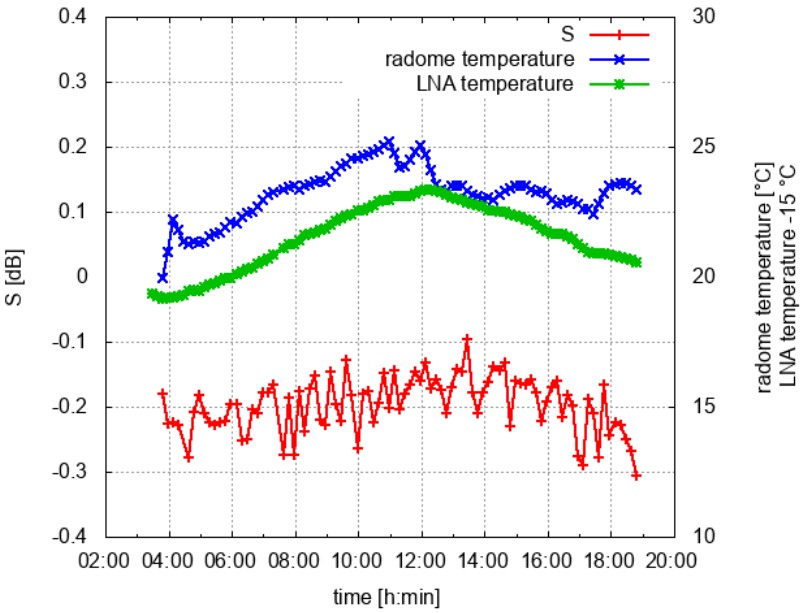

**Figure 7.** Variability of differential solar power S [dB] during the on 3 June 2018. Also shown is the radome temperature and the temperature near the LNA in the receiver box.

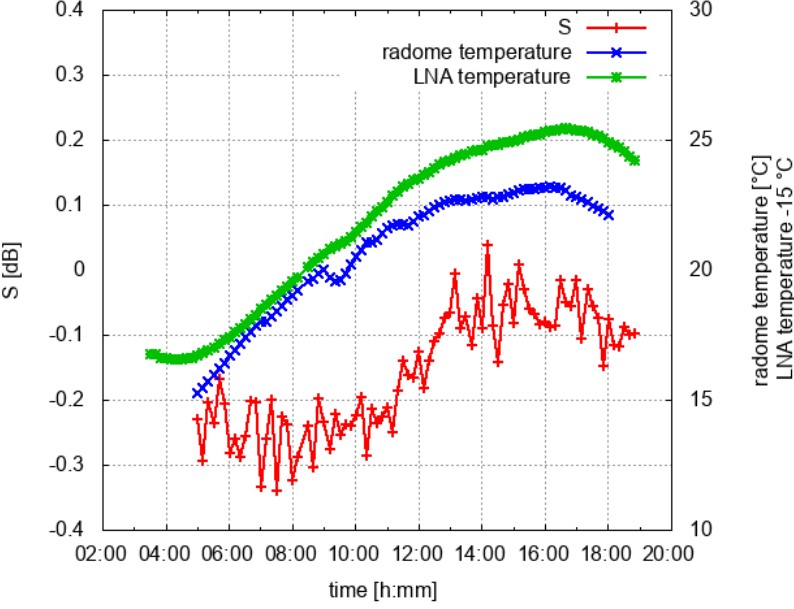

**Figure 8.** Variability of differential solar power S [dB] during the on 15 June 2018. Also shown is the radome temperature and the temperature near the LNA in the receiver box.





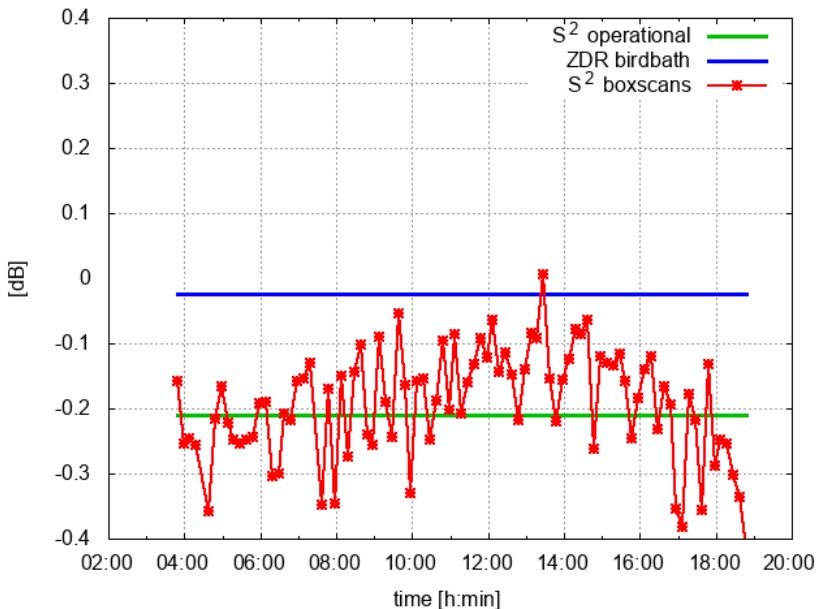

**Figure 9.** $S^2$ variability during 3 June 2018 compared to the $S^2$ and $Z_{DR}$ from operational monitoring. Operational data are updated once a day. No update of $Z_{DR}$ is possible if there has not been sufficient precipitation over the radar site on the previous day. In that case we keep the $Z_{DR}$ from the last precipitation event.

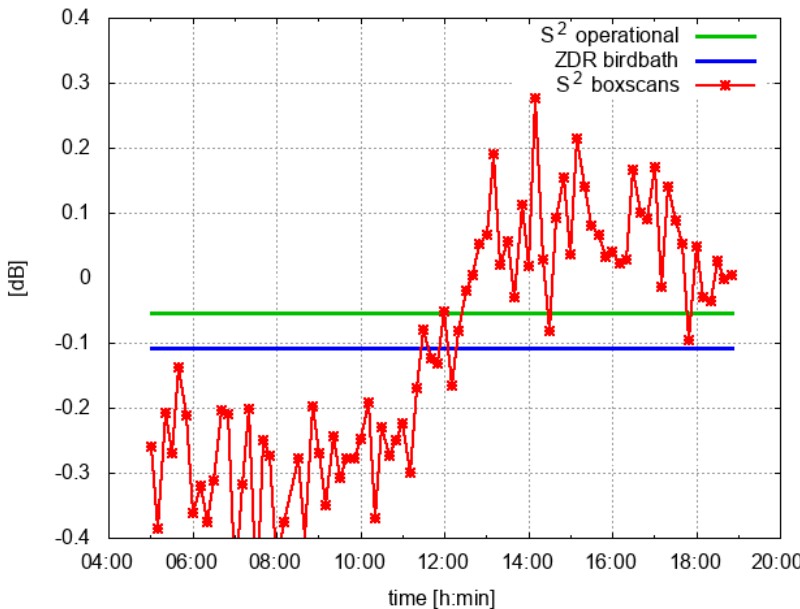

**Figure 10.** $S^2$ variability during 15 June 2018 compared to the $S^2$ and $Z_{DR}$ derived from operational monitoring. Operational data are updated once a day. No update of $Z_{DR}$ is possible if there has not been sufficient precipitation over the radar site on the previous day. In that case we keep the $Z_{DR}$ from the last precipitation event.





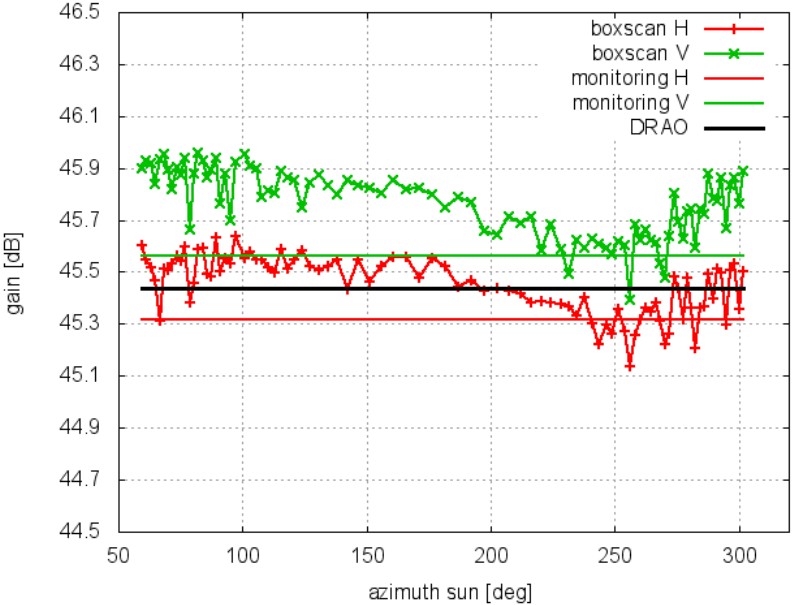

**Figure 11.** Antenna gain timeseries (as a function solar azimuth position) in H and V during 3 June 2018. Shown is also the antenna gain estimate from operational monitoring ("monitoring H", monitoring "V") and the gain estimate using DRAO data only.

boxscans) in Figs. 11 and 12 are compared to the gain estimate using the solar power derived from the operational monitoring and the gain estimate based on DRAO solar power data, for which there is just one value per day which is indicated the straight lines in the Figs. 11 and 12. For both the 3 June and 15 June 2018 data, larger variability in gain on the order of 0.2 dB is observed in the morning and evening when the elevation of the sun is low. Most likely surface clutter is the prime contribution

to this variation in gain. A decrease of gain on the order of 0.3 dB is observed during the day, followed by an increase of gain of about 0.3 dB in the evening hours.

The antenna gain provided by the manufacturer is 45.4 dB in H and 45.2 dB in V. The average gain based on the boxscan data in comparison to the gain based on the operational monitoring of solar hits are summarized in Table 1. Based on those measurements, there is a very good agreement between the gain estimates of H using solar power measured by radar and the

manufacturer value of 45.4 dB. Therefore, the assumptions on the antenna characterstics appear reasonable The deviation is within less than 0.2 dB (considering the gain in V). The gain estimates from the solar boxscans is about 0.3-0.4 dB larger in V than the estimate in H, and the manufacturer gain. This suggests a receiver calibration bias on this order which indicates that the receive channel actually is more sensitive (means that dBm0 should smaller by 0.3 - 0.4 dB).

Qualitatively, the diurnal variation of gain seems to suggest that there is a temperature correlation with a decrease in gain

during the day as a function of temperature, with an increase of gain in the end of the day when temperature deceases again (compare with the temperature variation for the two days shown in Fig. 13). This will be further investigated after analyzing the temperature dependence for the whole data set.



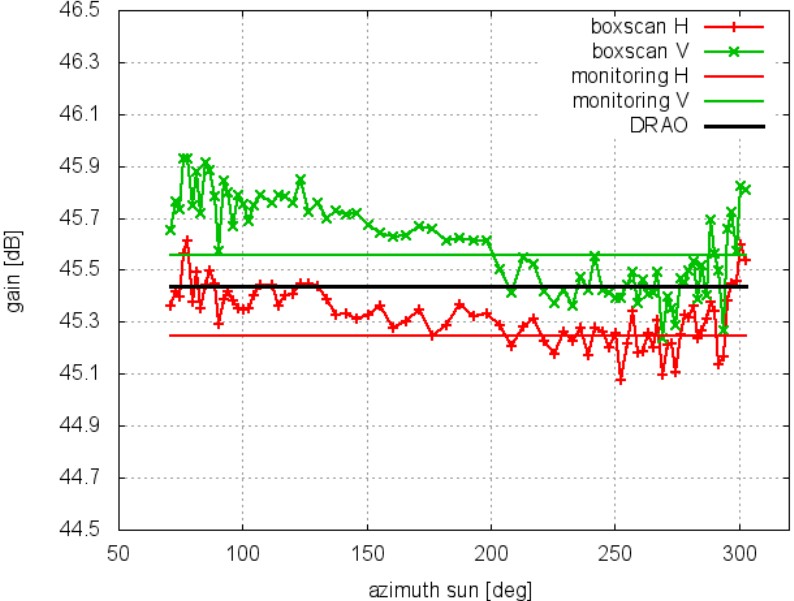

**Figure 12.** Antenna gain timeseries (as a function solar azimuth position) in H and V during 15 June 2018. Shown is also the antenna gain estimate from operational monitoring ("monitoring H", monitoring "V") and the gain estimate using DRAO data only.

**Table 1.** Mean antenna gain from boxscan data compared to the gain obtained from the operational monitoring. Antenna manufacturer gain is 45.4 dB (H) and 45.2 (V). The DRAO data based gain estimate is 45.4 dB.

| gain | boxscan mean | operational |
|------|--------------|-------------|
| $g_H$ (03.6.2018) | 45.4 | 45.3 |
| $g_H$ (15.6.2018) | 45.3 | 45.2 |
| $g_V$ (03.6.2018) | 45.8 | 45.6 |
| $g_V$ (15.6.2018) | 45.6 | 45.6 |

## 7 Temperature dependence of differential solar power $S$ and gain $g$

The $S$ variability as a function of temperature is evaluated based on over 2157 solar box scans acquired in early summer 2018. The temperature references for $S$ the radome temperature and the receiver temperature is taken. The radome temperature is the temperature in the volume surrounded by the radome. There is constant ventilation so that well mixed temperature conditions 5 can be expected. The RX temperature is measured in the receiver box close to the LNA which is mounted on a solid metal plate.

The scatter plots of $S$ versus radome temperature and LNA temperature are shown in Fig. 13. A temperature span of 12°C is captured with this data set. It is apparent that the correlation of $S$ and LNA temperature is better than the correlation





with the radome temperature especially for radome temperatures between 22 and 24°C. The bin-wise median of $S(T)$ (bin width is 1°C) together with the first and third quartile are shown Fig. 13 with black lines. This implies that origin of the $S$ temperature dependence is maybe located in the receiver box, where the analog components for H and V obviously have a differential temperature dependence, though the correlation does not show cause and effect. This is examined more closely

later. In the lower panel of Fig. 13 we fit a second-order polynomial $S_{fit}(T) = a \cdot T^2 + b \cdot T + c$ to the - RX temperature data, with $a = 0.00208 \pm 0.00002$ (dB / $T^2$), $b = -0.132 \pm 0.0152$ (dB / T) and $c = 1.86 \pm 0.27$ (dB). The fit to the curve resulting from the bin-averaged data is quite good.

Similarly, Fig. 14 shows the dependence of solar measured H and V gains on the receiver and radome temperatures. There is an obvious correlation of H and V gains with temperature. Best correlation appears if taking the receiver gain temperature as a

reference (Figure 14). If we use the bin-wise averaged gain for H and V to compute the differential solar power $S$ as $g_H - g_V$ we recover the curve shown in Fig. 13. The bin-wise averaged gain and the resulting is shown in Fig. 15. We note the gain decreases by about 0.6 dB over a temperature difference of 10°C and this indicates a temperature dependent bias of the radar reflectivity factor $Z$. This temperature dependence is an additional contribution to the overall uncertainty of calibration, which is usually not considered in error assessments from manufacturers, where commonly engineering uncertainties in measuring

the antenna gain, the transmit and receive losses among others, are considered.

The differential power and the gain temperature dependence appears to correlate best with the receiver temperature and thus the LNA's and associated circuitry could be responsible for the seen temperature dependent S observations. In order to evaluate the LNA gains, one-point calibration measurements are employed. Operationally, this test is carried out twice a day in order to monitor the calibration of the radar without an automatic adjustment of the radar calibration. With the one-point calibration,

a test signal is injected into the receive path using a built-in test signal generator. Since the response of the digital receiver is linear, the receiver response for a test signal above noise level and below the receiver saturation can be used to determine dBZ0 and dBm0. In order to investigate the temperature sensitivity, the one-point calibration was done every 5 minutes as part of the operational 5 minute scan cycle. At the the Hohenpeißenberg radar, a test signal can be coupled into the H and V receive path at the antenna crossguide coupler or at the LNA inside the receiver box. A 3dB power splitter provides nearly

equal power levels for both channels. This small power difference in inconsequential for evaluating the temperature sensitivity of the differential gain of the LNAs. The H and V signal path from the antenna crossguide coupler includes a waveguide filter, a circulator and a TR-limiter, respectively, which are all located behind the antenna outside the antenna mounted receiver box and will also manifest some dependence on temperature. The temperature stability of the built-in TSG was characterized in a climate chamber where a very small temperature dependent power output of the TSG of $\pm 0.05$ dB within a temperature range

between 0°C and 30°C was found. For proper calibration results, TSG losses between the coupler and the reference plane 1 (see Fig. 1) have been quantified using a network analyzer. The TSG losses between antenna coupler and the reference plane 1 are measured as 77.96 dB (H) and 77.89 dB (V). If the TSG signal is injected at the LNA, the losses have been determined as 26.16 dB for both channels. Any observed temperature sensitivity based on the one-point calibration therefore is attributed to the respective receive paths. A data set with 2562 calibration results, where the TSG signal was coupled into the antenna



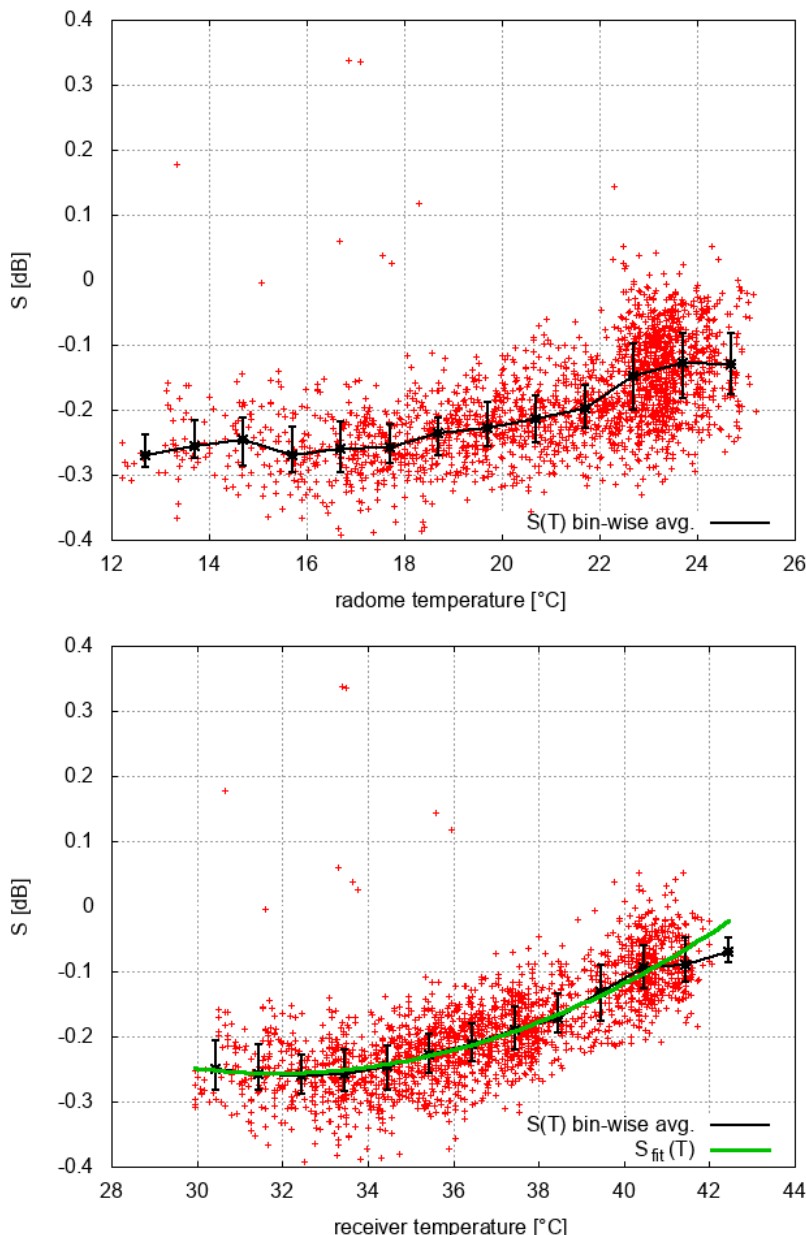

**Figure 13.** $S$ scatter plot as a function of radome temperature (upper panel) and LNA temperature (lower panel). LNA temperature is a good proxy for the temperature variations with in the receiver box. Shown is also the bin-wise averaged ($S(T)$ bin-wise avg; 1°C width and the 1st and 3rd quartile of all values within this bin), and a polynomial fit of second order ($S_{fit}(T)$, see also text) of $S$ versus LNA temperature (lower panel). In total 2157 boxscans are used for this analysis.



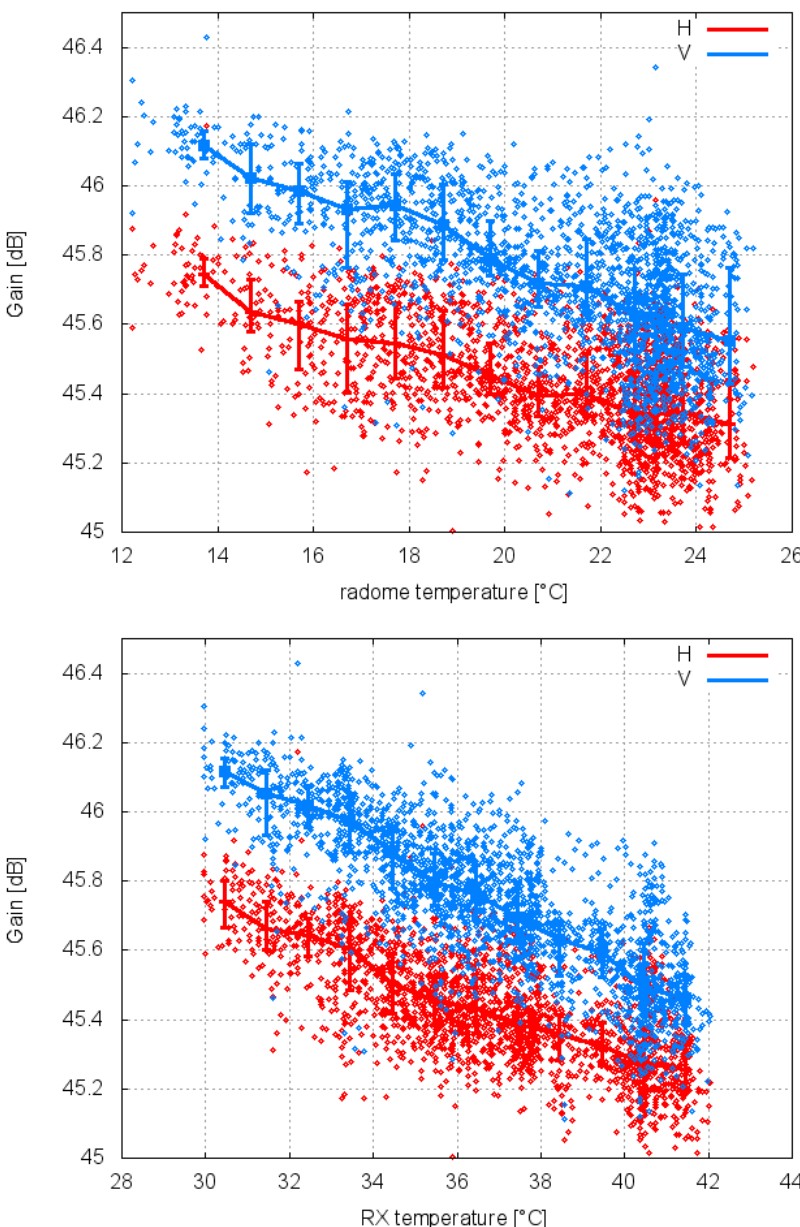

**Figure 14.** Gain (H & V) scatter plot as a function of radome temperature (upper panel) and LNA temperature (lower panel). LNA temperature is a good proxy for the temperature variations with in the receiver box. Shown is also the bin-wise averaged gain (1 C width and the 1st and 3rd quartile of all gain values within this bin) as a function of temperature. In total 2157 boxscans are used for this analysis.





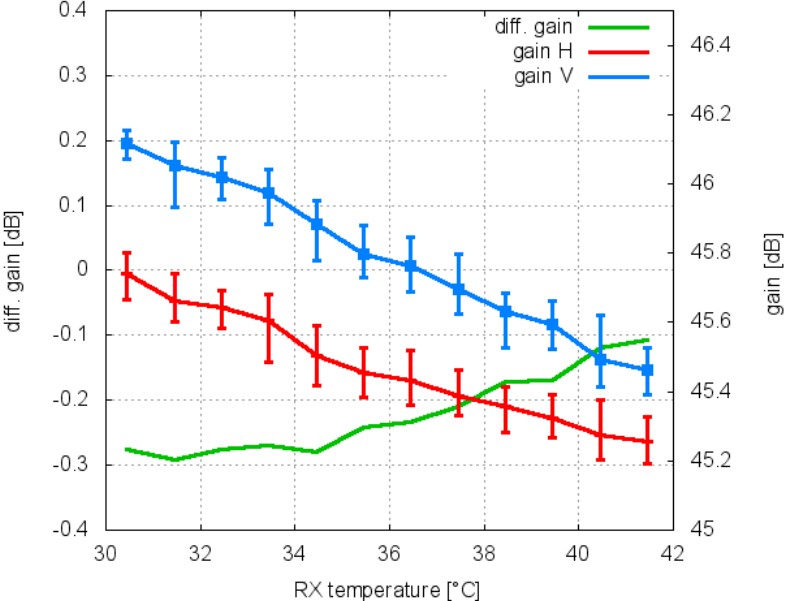

**Figure 15.** Bin-wise averaged gain (H & V, see Fig. 14) and computed as $g_H - g_V$ versus receiver temperature.

crossguide coupler, was acquired between 14 March 2019 and 23 March 2019. A second data set with 2810 calibration results, where the TSG signal was coupled into the LNAs, was acquired between 23 March and 5 April 2019.

The temperature dependent measured power samples are shown in Fig. 16. The temperature dependence for both data sets is identical if either the receiver temperature or the radome temperature is used. Similar to the retrieved gain based on solar

power, we find a 0.6 dB decrease over a comparable temperature range. The lower measured power levels in the lower panel of Fig. 16 are due to additional losses in the receive path. A remarkable feature of both independent data sets is the non-linearity of the horizontal power sample at around 7°C (radome temperature) or 19°C (receiver temperature) with a ≈ 0.1 dB dip. This is not found for the power sample in V. The reason for this is still unknown. The corresponding differential power H-V is shown in Fig. 17. The non-linearity in H leads to a -0.1 dB bias in differential power just for this specific temperature range.

For the the rest of the temperature range we find only a very small temperature dependence in differential power up to 0.02 dB over a range of ≈ 16°C. The differential power based on power samples where the TSG is coupled to the antenna crossguide coupler is on the order of 0.2 dB. This significantly larger difference compared to the other data set can be attributed to different insertion losses of the TR-limiters and the circulators, but also to the uncertainty of the measured TSG losses. But more importantly, there is again a very small temperature dependence of differential power.

This result is now compared to the $S$ temperature dependence shown in Fig. 13. Even though the temperature ranges do not match precisely, some conclusions can be drawn. Based on solar boxscan data a 0.2 dB increase in $S$ over a temperature range of 12°C is found. The correlation with temperature is best if the receiver temperature is used to determine the mean temperature dependence of $S$. In contrast, the temperature sensitivity of differential power based on the one-point calibration

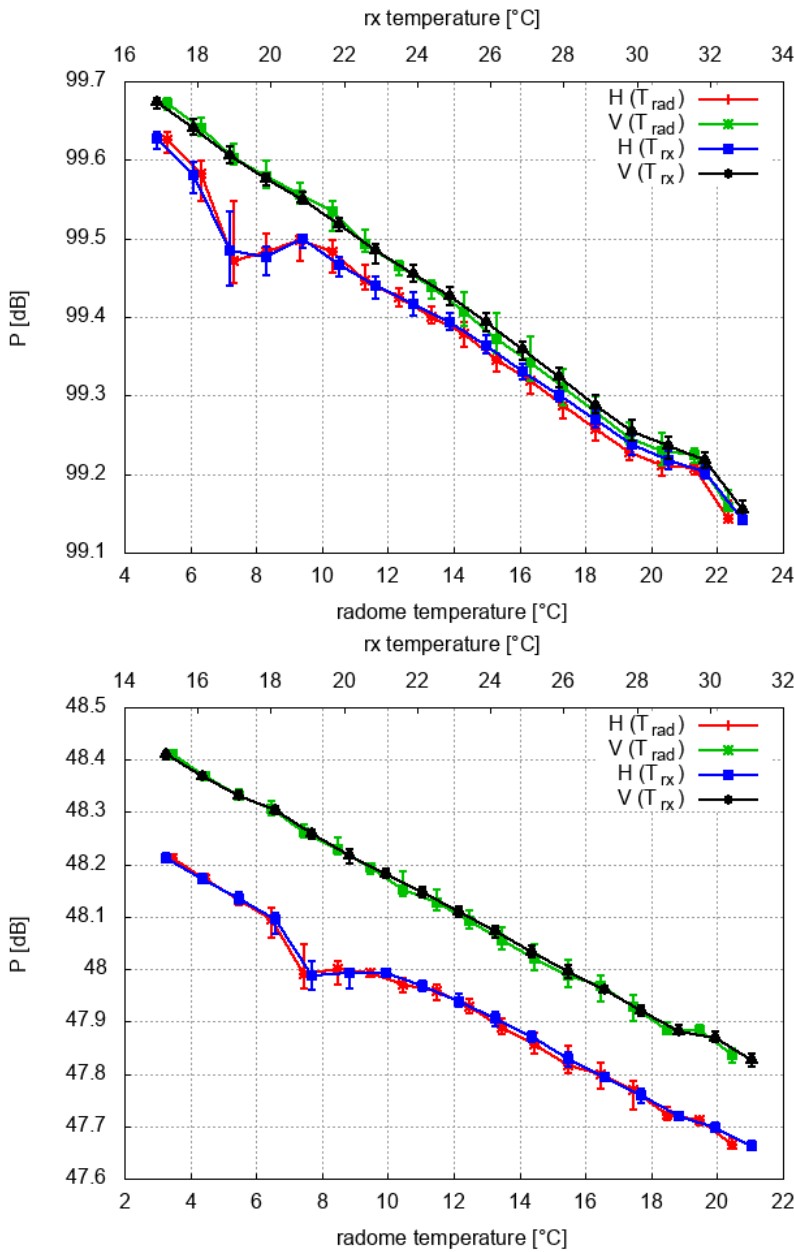

**Figure 16.** Bin-wise averaged H and V power measurements based on the one-point calibration. Either the radome ($T_{rad}$) or the receiver box temperature ($T_{rx}$) is used as a reference. Shown is the median, the first and third quartile of 1°C wide temperature bins. The upper panel shows the results with the TSG coupled to LNA, and the lower panel coupled to the antenna crossguide coupler.

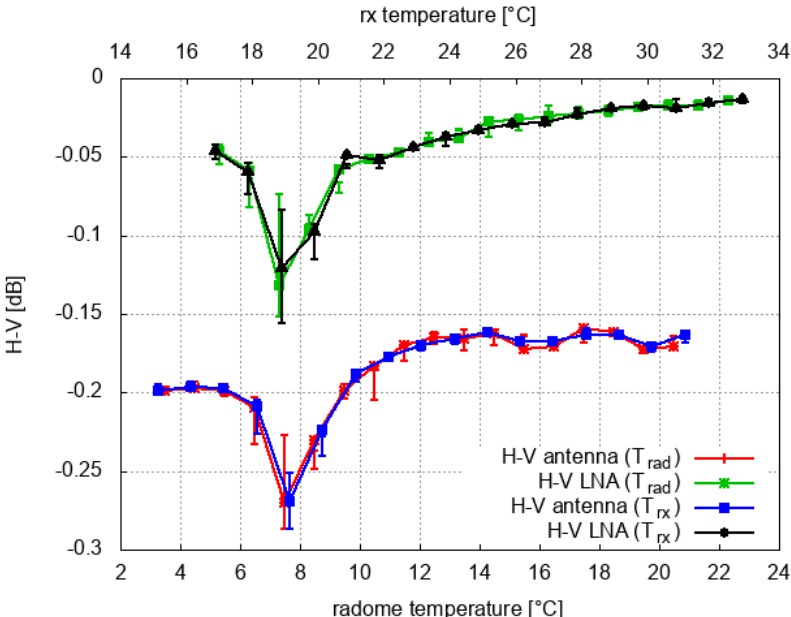

**Figure 17.** Bin-wise averaged differential power H-V based on the power samples from the one-point calibration. Either the radome ($T_{rad}$) or the receiver box temperature ($T_{rx}$) is used as a reference. Shown is the median, the first and third quartile of 1°C wide temperature bins.

indicates a substantially smaller temperature dependence with an increase of 0.02 dB or less over a comparable temperature range (excluding the anomalous behavior at 7°C/19°C). This behavior is essentially identical if either the radome temperature or the receiver temperature is used as a reference for the one-point calibration results. Therefore the temperature sensitivity observed in $S$ can be attributed to the antenna assembly. [2] This is in agreement with the results in Hubbert (2017) who arrived

5  at a similar conclusion. It is argued, that the thermal expansion of the antenna assembly (including the struts) is responsible for large part of the temperature sensitivity of differential power. This is still a matter of investigation, but initial simulations of antenna radiation patterns appear to support this conclusion.

## 8   Longterm $Z_{DR}$ monitoring in the DWD weather radar network

Time series of the $Z_{DR}$ calibration are available since the beginning of 2013. From the beginning $Z_{DR}$ was monitored using the

10  birdbath scan and the monitoring using solar interferences. How the $Z_{DR}$ values agree for specific days has been shown in the previous section. The success of the automated $Z_{DR}$ calibration procedure is discussed for all 17 radar sites up to September 2018 (Fig. 18). The methodology to determine $Z_{DR}$ is described in section 2. In Fig. 18 each data point represents a diurnal averaged $Z_{DR}$ value, which can only be determined if there are at least 6 birdbath scans available with precipitation. In total

---

[2]There is one caveat however. There is a chance that the temperature dependence may be different in the two compared temperature ranges. This will will be investigated in future measurements.





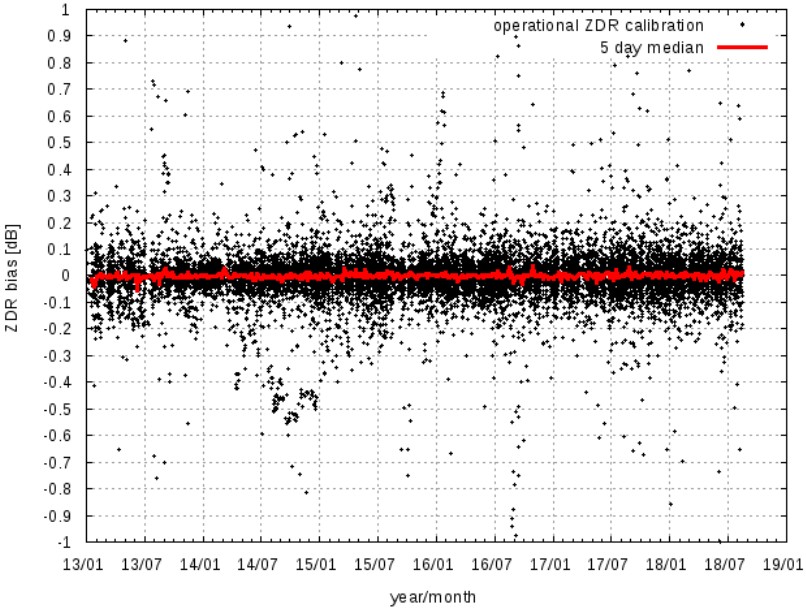

**Figure 18.** Radar network wide calibrated $Z_{DR}$ using birdbath data. Also shown is the median $Z_{DR}$ over a 5 day interval from January 2013 until September 2018). There is no systematic annual variation of $Z_{DR}$ which implies that it is not necessary to distinguish into the precipitation phase when calculating the $Z_{DR}$ from birdbath data.

16646 dirunal averaged $Z_{DR}$ values (which corresponds to 45 years of data) are shown in Fig. 18. The corresponding total radar operation time amounts to 31928 days (about 87 years). There is on average sufficient precipitation present on 52% of the days. So for a given day, there is precipitation over the site for at least six birdbath scans. The median $Z_{DR}$ is 0.0 dB, and the mean absolute deviation MAD is 0.029 dB (Wilks, 2011, Frech et al, 2017). This indicates that the method to update the $Z_{DR}$

offset once a day appears sufficiently robust to provide a well calibrated $Z_{DR}$. Further optimization is needed to eliminate the apparent outliers which may be caused by e.g. unwanted clutter contributions. Note that the diurnal variability of $Z_{DR}$ due to e.g.temperature, as discussed in this paper, is not captured by this automated procedure to determine the $Z_{DR}$ offset. From one scan to another there might be deviations larger 0.1 dB due to e.g. temperature effects. The existence of larger $Z_{DR}$ deviations are also indicated by the data in Fig. 18. In addition, the median $Z_{DR}$ in non-overlapping 5 day intervals is computed

using the calibrated $Z_{DR}$ data from the whole network. There appears no annual variation in $Z_{DR}$ bias (Fig. 18) which is an indication that the method can be applied as a season independent method without restricting the $Z_{DR}$ offset analysis to liquid precipitation only.

    Note, that the radar systems have similar performance with respect to $Z_{DR}$ bias (Fig. 19). There is only one system (radar Neuhaus, id 9) which sticks out. But even this system is within the target corridor of $\pm 0.1$ dB.

The automated adjustment is well suited to correct for system drifts on a timescale larger a day. This is shown in Fig. 20 for the Eisberg system. This is the case of an unusual TR-limiter degradation of the Eisberg radar. TR-limiters are usually





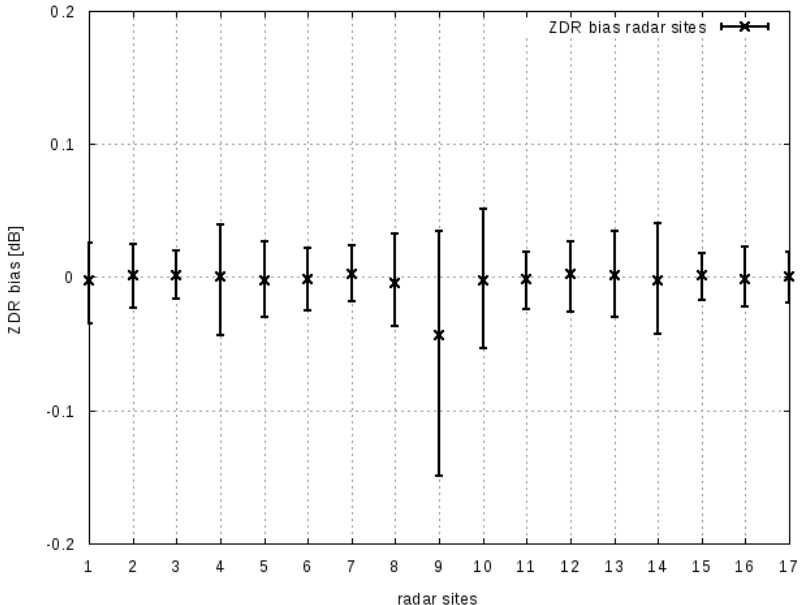

**Figure 19.** $Z_{DR}$ bias of the 17 DWD radar site January 2013 - August 2018. Shown is the median $Z_{DR}$ and the 1st and 3rd quartile.

located in from of the LNAs in order to protect the receiver from the transmit pulse (see Fig. 1). The transmit pulse causes the TR-limiter to act as an open circuit for a short period of time called the recovery time. Typically an aging TR-limter shows an increase of its recovery time which causes an undesired attenuation of the received signal at ranges at first close to the radar but then extending in range as the TR-limiter fails. Here, the overall duration of attenuation of TR-Limiter increased with time,

which is illustrated by the fact that the $S$ has a similar drift like the $Z_{DR}$ determined from the birdbath scan. Over a time period of 9 months there is an increase of $Z_{DR}$ bias of 3 dB. Since the increase in attenuation happens to be on a timescale larger than one day, the operational $Z_{DR}$ offset adjustment is able keep $Z_{DR}$ within 0.2 dB ("$Z_{DR}$ calibrated 90°", Fig. 20). After the replacement of the faulty TR-limiter, the system operated reliably for about 3 months before the other limiter started to degrade in a similar way as the other limiter. Within about 9 months the $Z_{DR}$ bias increased to -4 dB. The $Z_{DR}$ bias is

constant after the replacement of the TR-limiter in October 2016. On the scale shown here, the Zdr bias as estimated from the birdbath measurement ie well approximated by $S$. Since $S$ is only a function of the receiver electronics and the antenna, and the birdbath measurements are a function of the receiver electronic, the antenna gains squared and the transmit power, (Eqs.(6) and (11) respectively), it can be concluded that the transmit power ratio at the reference plane is relatively constant. Diurnal variations of $S$ due to temperature are not seen in this daily average data.

Similar cases have been observed at other radar sites. The reason for this type of TR-limiter failure is not known yet. This example nicely illustrates the benefit to use and combine different data sources to monitor and calibrate $Z_{DR}$.

Considering the classic aging of a TR-limiter, a prototype of an automated analysis of the shape, and in particular the slope of the $Z_{DR}$ profile has been implemented. Results are currently analyzed. This approach will provide the height interval over

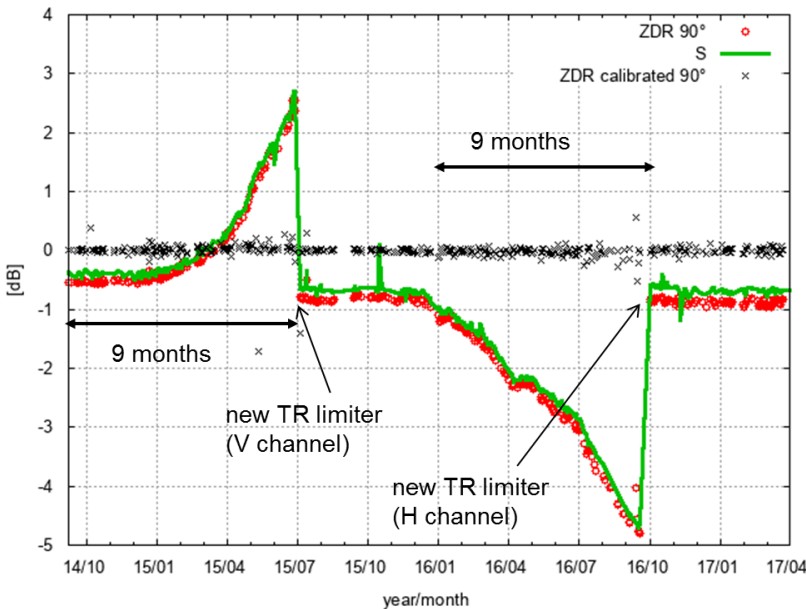

**Figure 20.** TR-limitier failure at the Eisberg radar. This illustrate the benefits to use solar monitoring and birdbath scan to assess biases in the receive path. Here, the combination of solar monitoring (solar power $S$) and birdbath analysis isolates the receive path as the source of the $Z_{DR}$ problem. In addition we show the resulting calibrated $Z_{DR}$, where the calibration offset is determined from birdbath data automatically.

which the mean $Z_{DR}$ should be computed from the birdbath profile. Another promising approach to monitor the classic aging of a TR-limiter is the monitoring of clutter power in close vicinity of the radar (Mathijssen et al, 2018).

## 9 Examples of antenna gain retrievals from the radar network

Full diurnal cycles of solar boxscans have been acquired from four operational identical radar systems: Flechtdorf, FLD
5 (27.6.2018); Neuheilenbach, NHB (24 July 2018); Boostedt, BOO (27 June 2018); Hannover HNR (24.7.2018). The solar boxscans were evaluated similarly to the Hohenpeißenberg boxscans. The resulting diurnal series of gain estimates for H and V as a function of the solar azimuth position are shown in Fig. 21. In addition, the MHP gain estimates from 3 June 2018 are shown for comparison. Aside the Flechtdort $g_h$, all gain estimates for both polarizations are within a 1 dB range and close to the nominal antenna gain of 45.4 dB. The diurnal variation of BOO show a decrease in gain of up to 0.2 dB around a solar
10 azimuth of 90° and 270°. The decrease is more pronounced for the vertical than the horizontal polarization. This feature can be related to the presence lightning poles outside the radome. This decrease is not visible for the other operational systems because the azimuthal position the lightning poles is such that elevation of the sun during the measurements is too high so that the poles were not in the field of view of the antenna. Similar to MHP, a slight decrease of gain in the course of a day by 0.2 dB is also found for systems HNR, FLD and BOO. Diurnal temperature variations are the likely cause for this. The nominal





antenna gain based on antenna pattern measurements is about 45.4 dB. A retrieved gain value $g_H$ of close to 47 dB from FLD indicates a calibration error of 1.6 dB which can be attributed to presumably a erroneous characterization of the horizontal receive path.

## 10    Summary and Conclusions

Various practical aspects of calibrating and monitoring the calibration state of the DWD radars were investigated in this paper. Measurements of solar H and V powers are particularly useful in this endeavor since the solar radiation at C band can be considered unpolarized and thus H and V solar powers incident on the radar antenna are equal. It follows that any differences from 0 dB in the ratio of the measured H and V solar powers ($S$, Eq.(11)) indicates a $Z_{DR}$ bias caused by the radar's total receive path (antenna plus receiver hardware and electronics). Variations of $S$ a a function of time provide insights into the sources of the previously observed temperature sensitivity of $Z_{DR}$ bias within the DWD weather radar network. To this end, over 2000 dedicated solar boxscans measured with the DWD dualpol C-band research radar Hohenpeißenberg were used to investigate the variability of $S$. The Hohenpeißenberg radar system is identical to the 17 radar systems of the DWD network and is operated 24/7 like an operational system if it is not being used for research purposes. The solar boxscans were complemented with the analysis of differential power data from one-point calibration measurements, which were carried out every five minutes as part of the operational five-minute scan cycle in spring 2019. Using a built-in TSG, test signals were coupled into either the antenna crossguide coupler outside the receiver box at plane 2 (see Fig. 1) or into the LNAs inside the receiver box. Prior those tests the temperature dependence of the power output of the TSG was assessed in a climate chamber. Over a 30°C temperature range, power variations smaller than 0.05 dB were observed. Based on the solar boxscans, a non-linear temperature dependence of solar differential power $S$ with a 0.2 dB increase in a temperature range between 30 and 40°C was found. Differential power measurements based on one-point calibration data indicate a temperature dependence of less than 0.02 dB over a comparable temperature range. This indicates that the $S$ temperature sensitivity can be attributed to the antenna assembly. This is consistent with the analysis of Hubbert (2017). Thermal expansion of the antenna assembly, which includes the struts, seem to cause a large majority of the observed temperature sensitivity of $S$. In order to avoid the observed $S$ variability, the antenna assembly should be kept at a constant temperature or a correction factor based on the average functional relationship between $S$ and $T$ could be applied.

Solar boxscans provide a simple and straight forward approach to assess and monitor antenna characteristics of a weather radar network. It is suggested that solar boxscans can be used to monitor the antenna assembly throughout the lifetime of a radar system if the differential power variability of the receive electronic can be kept to a minimum (say, 0.05 dB). With the DWD radar operation software, solar boxscans can be scheduled remotely like an operational scan. Antenna beam widths derived from the solar boxscans scans also assist in detecting the degradation of antenna assembly. If the feed would be out of focus [3], a mismatch of the H and V beam could be discerned. The solar $S$ antenna pattern agreed well with the $Z_{DR}$ antenna pattern that was measured during a standard antenna pattern measurement in 2011 during the installation phase of the radar

---

[3]The phase center of the feed horn should be aligned with the focus of the parabola.

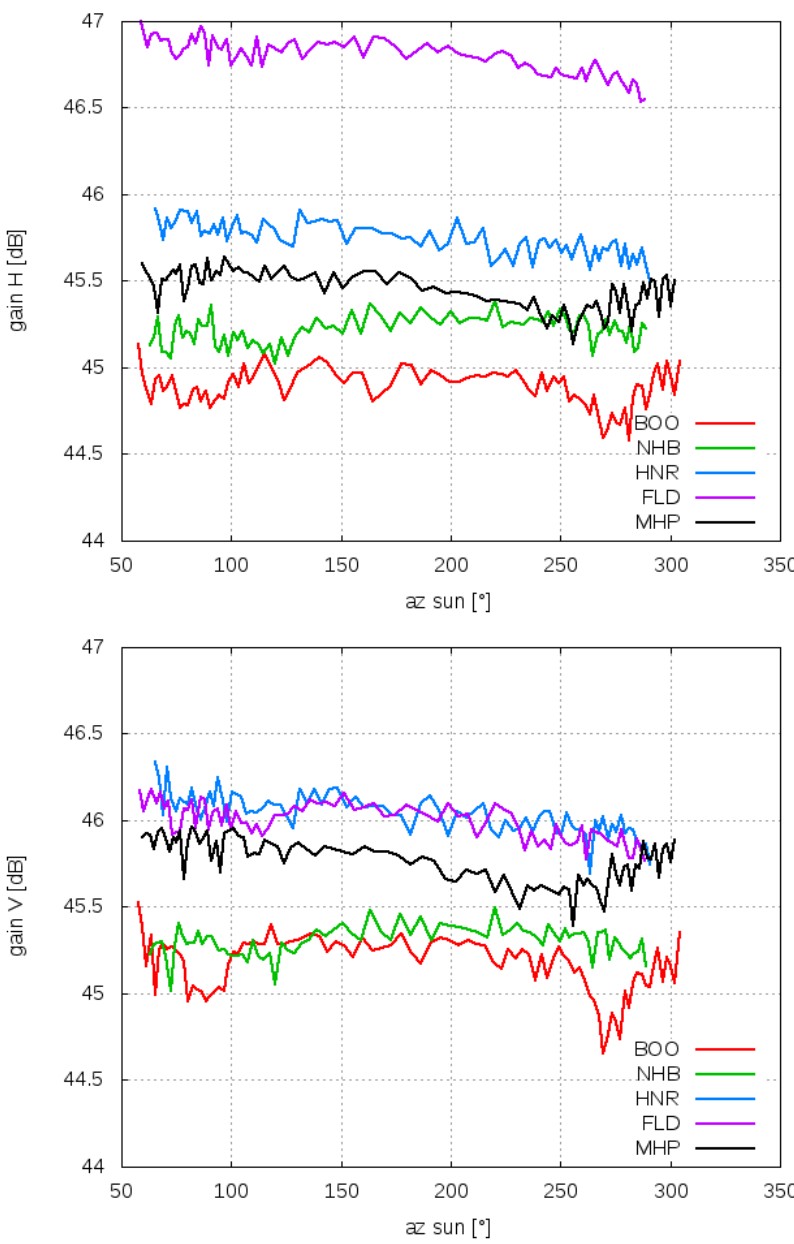

**Figure 21.** Diurnal gain estimates as a function of solar azimuth from five radar systems, Boostedt (BOO), Flechtdorf (FLD), Hannover (HNR), Neuheilenbach (NHB) and Hohenpeißenberg (MHP). Shown is the gain of the horizontal receive channel (top panel) and vertical receive chanel (bottom panel). The timeseries shown here are based on about 90 box scans that are available for each station.



system. It was shown for the Hohenpeißenberg that the differential H to V antenna pattern from the standard antenna pattern measurement matched well the $S$ pseudo antenna pattern. The main H beam shape was circular, whereas the main V beam shape was slightly elliptical.

The antenna gain was estimated using the radar measured solar power and the solar power based on the solar flux measurements of the Dominion Radio Astrophysical Observatory (DRAO) in Canada. If the receive path is properly characterized and calibrated, the retrieved antenna gain should match the antenna gain as provided by the antenna manufacturer. A systematic deviation is then indicative of a receiver characterization bias, and trends in gain may reflect a temperature influence related to the receive path. Not surprising, a differential temperature dependence of gain was found. The temperature sensitivity of the H and V gain was a linear decrease of 0.6 dB over a 10°C temperature range, which directly relates to a radar reflectivity Z bias of 0.6 dB. Data from one-point-calibration measurements revealed a similar decrease in H and V gains. Thus, the temperature dependent H ands V gains can be directly related to the receiver electronic path. This contribution is typically not considered in the error budget of the radar equation and should be viewed as significant if the common target accuracy of ± 1 dBZ is desired. Based on two case studies, the antenna gain based on the measured solar power is within 0.2 dB to the antenna gain provided by the antenna manufacturer. There is also very good agreement, within 0.2 dB, between the retrieved gain estimates based on operational solar monitoring and solar box scans.

A full diurnal cycle of solar boxscans from four operational radar sites were compared to the Hohenpeißenberg data. With respect to gain all sites are roughly within 1 dB in H and V. There is only one site where the retrieved gain indicated a bias of about 1.5 dB which points to a calibration problem.

Results of the longterm operational $Z_{DR}$ calibration based on operational birdbath scans from the DWD radar network were given. It was shown that $Z_{DR}$ biases, which may occur on a timescale larger than 1 day, can be automatically corrected based on birdbath $Z_{DR}$ data. The analysis is based on over 87 years worth of radar data from the DWD radar network. A specific case with an unusual degradation of the two TR-limiters during a time span of 2.5 years was shown. Even though significant $Z_{DR}$ bias was present, the $Z_{DR}$ calibration procedure was able to keep the $Z_{DR}$ bias within ±0.1dB. This is because the increase in bias was slow and steady, on a timescale larger than one day. Using $S$ data from the operational solar monitoring, the receive path could be identified as the source of the bias. Note, that this is a rather unusual TR-limiter behavior compared to a typical degradation of TR-limiter, where the recovery time usually increases, since the overall attenuation increased substantially. Within nine months, the TR-limiter attenuation increased up to 4 dB.

## Appendix A: Data

Table A1 gives an overview on the dates where typically full diurnal cycles of solar boxscans where acquired.

*Acknowledgements.* We acknowledge the fruitful discussions with the DWD radar team, in particular with Jörg Seltmann, Theo Mammen, Kay Desler, Norbert Engler, Bertram Lange and Benjamin Rohrdantz.



**Table A1.** Overview on solarbox scan database used in this work. In total 2157 solar boxscans are used in the analysis.

| date | number of boxscans |
|---|---|
| 15.4.2018 | 71 |
| 17.4.2018 | 76 |
| 18.4.2018 | 79 |
| 19.4.2018 | 77 |
| 20.4.2018 | 81 |
| 21.4.2018 | 66 |
| 22.4.2018 | 81 |
| 27.4.2018 | 83 |
| 07.5.2018 | 84 |
| 08.5.2018 | 84 |
| 03.6.2018 | 92 |
| 10.6.2018 | 92 |
| 11.6.2018 | 47 |
| 15.6.2018 | 92 |
| 16.6.2018 | 93 |
| 19.6.2018 | 90 |
| 20.6.2018 | 90 |
| 22.6.2018 | 94 |
| 23.6.2018 | 91 |
| 27.6.2018 | 90 |
| 30.6.2018 | 79 |
| 01.7.2018 | 90 |
| 02.7.2018 | 91 |
| 07.7.2018 | 75 |
| 08.7.2018 | 90 |
| 09.7.2018 | 88 |

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
