# Peer review of "Monitoring the differential reflectivity and receiver calibration for the German polarimetric weather radar network"

_Atmospheric Measurement Techniques, 2019_

## Referee Comment (RC1) · Richard Ice (Referee) · 26 Sep 2019

on lines 22 and 23, the authors describe the engineering calibration method as "untenable" for NEXRAD, the US Doppler network based on the WSR-88D. This does not accurately represent the current situation. The US program relies solely on the aforementioned engineering methods, combined with solar scans, to set the calibration point for differential reflectivity. The calibration state is then monitored using external targets, rain, snow, and Bragg scatter, along with solar interference's or sunspikes. These methods assess the state of calibration but do not correct it. Radars that are assessed to be beyond acceptable limits are addressed on a case by case basis.

[Figure]

At this time, in any given month, approximately 70 percent of the network sites are assessed to be within acceptable limits of + or - 0.2 dB, with many of them performing better. It is possible the sites that are found to be outside these limits are subject to the temperature dependencies documented in this paper.

Recommend the authors change the term "untenable" to "challenging".

Overall, this an excellent paper and it addresses critically important results.

---

## Referee Comment (RC2) · Martin Hagen (Referee) · 10 Oct 2019

The manuscript documents the monitoring of the ZDR and receiver calibration of the German weather radar systems in great detail. A temperature dependent bias has been found and attributed to the antenna assembly. The paper is well written and deserves publication within AMT after some minor modifications.

General remarks:

The manuscript is quite elongate and could be shortened removing details which are not relevant or which are repeated a few times. On the other hand it gives sufficient

details to follow the various steps of analysis. Considering this it is worth to have more details than what would be necessary for a pure scientific publication.

What I feel confusing is the fact that in the beginning it is suggested that the observed temperature dependence of the ZDR bias is caused by the receiver electronics, whereas the analysis shows that it seems to be related to the antenna assembly. In particular, the discussion of Fig. 13 does imply the electronics origin.

Minor remarks (probably there are more typos):

Page 2, line 5 and 10: give credit to Seliga & Bringi, 1976, they did some analysis on required ZDR accuracy

Page 3, line 3-6: partly repeated from page 2 line 15

Page 3, line 12: the sun is a source of electromagnetic radiation, not only in S-band

Page 5, line 3-5: repeated from page 3 line 3-6

Page 5, line 12: Bringi and Chandrasekar 2001  j

Page 6, line 7: there is no section 11

Page 6, line 8-16: this passage should go somewhere else, maybe to section 3.1

Page 6, line 27-30: the sentences about the three circulators could be omitted.

Page 6, line 32: the data format is not relevant for the paper

Page 9, line 4 and 13-15: keep the advantages of gain together

Page 9, line 13: ... in both *gain* estimates

Figure 2, 3, and 6: what is the meaning of the red and blue circles and + and x

Page 14, line 19: is there any possibility to see whether the contributions of the sun hits are equally distributed over the day or whether they are more frequent in the morning and evening hours (what would be my assumption for measurements in June)

Page 14, line 21: was there sufficient precipitation for the days prior to the days shown?

Page 14, line 24 and 25: the figures show about -0.2 and -0.05 for 3 June and 15 June, respectively

Page 17, line 2; ... is indicated *by* the straight lines

Page 17, line 10: ... appear reasonable *.* The deviation ...

Figure 11 and 12: why not time on x-axis? The other figures show time on x-axis

Page 19, line 2-7: the detailed analysis suggests that the electronics is the cause of the temperature dependence, but this is revoked later on

Page 19, line 11: ... and the resulting *??* is shown ...

Page 19, line 18: one-point calibration is better known as "single-point calibration"

Page 19, line 24: show insertion points at Fig. 1

Page 19, line 25: ... power difference *is* inconsequential for ...

Page 26, line 1: located in *front* of the LNAs . . .

Page 26, line 3: at ranges ??at first?? close to the radar

Page 27, line 3: ... similarly to the Hohenpeißenberg *, MHP* boxscans.

Page 28, line 19: since S is related to antenna temperature, the radome temperature range of 15 to 25°C should be given and not the one related to the LNAs

Figure 21: time on x-axis like in Fig. 7 and 8

Page 30, line 1: ... for the Hohenpeißenberg *radar* that the ...

Page 30, line 29: I guess that the dates are for the Hohenpeißenberg radar

---

## Short Comment (SC1) · 2 Nov 2019

Differential reflectivity (Zdr) is a polarization moment used for clutter removal, hydrometeor classification, and for quantitative precipitation estimation (QPE). It is well-known that to limit uncertainties in QPEs, Zdr accuracy must be better than 0.2 dB. The paper illustrates Zdr techniques routinely implemented in the German weather radar network to keep Zdr unbiased and it investigates diurnal Zdr variations related to environmental temperature. The work is valuable, and the topic is relevant in particular for operational weather radar where data quality techniques are limited by continuous operational scans. The language is appropriate and clear.

[Figure]

Specific comments

The authors attribute Zdr variability according to environment temperature is due to the antenna assembly in agreement with Hubbert (2017) findings. The authors should add more technical information on DWD weather radar antenna assembly and add comments on their findings respect to Hubbert's study. In fact, he investigated the same topic on S-Pol weather radar with different antenna assembly. How are the current results related with Hubbert (2017)? Do completely different antennas produce the same Zdr variations depending on temperature?

Technical comments

Page 3 line 10: Holleman et al. (2010), → Holleman et al., 2010, Page 3 line 34: This analysis is based on a combined 87 years of radar operation ? Page 5 line 5: , i.e., 360âŮ̊ę 5 azimuth → , i.e. 360âŮ̊ę 5 azimuth Page 5 line 13: and at least ten valid range bins in a ray. How long is a range bin? Page 5 line 21: $\Delta$TX,RX,t0 → $\Delta$TX,RX Page 9 line 17: Since basic antenna parameters are fixed, the gain estimates cannot be viewed as true antenna gain estimates. Could do the authors specify better? Page 9 line 8: eq. → Eq. Page 9 line 22: 3dB → 3 dB Page 12 Figures: daytimes are in UTC or local time? Please, specify. Page 14 line 6: a 5th order polynomial → a 5th order polynomial Page 18 Table 1: although quite obvious, please add units (dB) in Table header. Page 24 line 3: Therefore the temperature → Therefore, the temperature Page 24 line 4: please move footnote 2 in the text to increase readability. Page 24 line 9: of 2013 → of 2013 Page 26 line 10: he Zdr bias as estimated from the birdbath measurement ie well approximated by S. Please correct typing errors in the sentence. The authors sometimes report "Eq. X" and sometimes "Eq. (X)". Please correct Page 30 lime 30: than 1 day → than one day

───────────────

---

## Author Comment (AC1) · 3 Dec 2019

Thank's a lot for clarifying the current status and procedure on ZDR calibration for NEXRAD. We are now using the suggested wording.

---

## Author Comment (AC2) · 4 Dec 2019

We thank the reviewer for his helpful comments which helped to sharpen and clarify the paper.

With respect to the possible confusion regarding analysis on the source of the observed temperature dependence: The main findings are summarized in the abstract. So the overall picture is layed out there and the paper takes the reader step by step through the analysis steps which then eventually lead to the conclusion, that, consistent with analysis of Hubbert (2017), the temperature dependence of ZDR can be attributed to the antenna assembly.

[Figure]

Changes to the manuscript:

We have considered the minor remarks (grammar, typos and stylistic) when revising the paper. Some specific responses to some of the remarks:

p. 3, l 12: we have changed the statement as suggested.

p. 6, l 8-16: we moved this part to page 8, section 3.1.

p 6. l. 27-30: the section on the circulators has been removed.

Figure 2,3,6: question on the symbold: these are the estimated pointing biases (discussed in Frech et al, 2019) for the H and V polarization. That information is now given in the caption.

p.14, l 9: You mean the sunhits from operational scanning? This could be achieved by plotting the solar azimuth instead of time. Such a representation is more relevant if we want to compare the azimuth and elevation bias from operational scanning compared to elevation and azimuth bias from solar box scans. This is discussed in Frech et al., 2019

p. 14, l 21: the date with sufficient precipitation is now indicated in the captions: for the case study from 3 June (solar boxscans) the corresponding birdbath data are from 1 June. For the 53 June case, the birdbath data shown are from 13 June.

Figure 11 and 12: (time versus plotting azimuth sun): We have changed the x-axis and plot the gain retrieval versus the time, so we are consistent with the previous plots (instead of plotting the solar azimuth).

P19, l 2-7: we now make the intitial statement that the S temperature sensitivity actually can be attributed to the antenna assembly. With this we think the reader is better able to follow the line of data analysis.

P. 19, l 24: Fig 1 is a schematic picture with a focus on the reference planes. It should not reflect the all elements of the tx and rx path of our system. We now refer to the

calibration diagram of the DWD radar system in Frech et al 2017 (Figure 2 therein), where the location of the cross guide couplers in the rx/tx path are shown.

P 28, l. 19: we now use the radome temperature range as suggested.

Fig 21: Here we keep the solar azimuth because we compare data from different dates (separated over a month).

P. 30, l. 29: Yes this is all Hohenpeißenberg data. This information is now included in the caption and the text.

---

## Author Comment (AC3) · 4 Dec 2019

We thank the reviewer for his helpful and contructive comments.

With respect to additional technical information: we refer to the Frech et al. 2013 where details of the DWD radar system (including the antenna assembly) are introduced.

With respect to the question on the findings in Hubbert (2107):

Hubbert (2017) also showed a Zdr bias that depended on the temperature of the antenna assembly. This physically designates everything to the right of the reference plane in Fig. (1). For S-Pol this encompasses the waveguide out of a sea container, up

through the rotary joints, and then along the dish support struts and to the feed horn and the parabolic reflector. The part of the antenna assembly and the exact physical cause of the Zdr bias variation is still unknown but is under investigation. For S-Pol an extensive antenna modeling study is being pursued. Since there are no active electronic components in the antenna assemble, it would seem then that the expansion and contraction of the parabolic reflector and the support struts are the most likely causes of the Zdr bias variability. The DWD antenna assembly is much the same. For S-Pol we have found that the Zdr bias as a function of temperature is not linear and it is also a function of frequency (see Hubbert 2017).

Changes in the manuscript according to the technical comments by the reviewer:

We have taken care of all the typos and obvious errors as indicated by the reviewer:

Some specific changes to the manuscript and comments:

page 3, line 34: If you add up the radar operation time of all 17 operational radars that are are analyzed here you come up with 87 years of radar operation time.

page 5, line 13: range bins of the birdbath scan have a length of 25 m

page 9, line 17: It is not the intention to provide an alternative method to determine antenna gain (so you are correct that this is not a true gain estimate, which is also stated in the paper, see p. 9, l 17 of the submitted manuscript). But the retrieved gain using e.g. observed solar flux at C-band serves, as shown here, is a good estimate of the antenna provided by the antenna manufacturer. The advantage to use the retrieved gain as compared to retrieved solar flux units (as it is commonly done in literature) is stated in this section of the paper.

page 12 figures: Where timeseries are shown we now indicate that time is in UTC.

page 24, l4: footnote is now in the text

We now use Eq. (X) throughout the text.